# STATE CHRONO REPRESENTATION FOR ENHANCING GENERALIZATION IN REINFORCEMENT LEARNING

## ABSTRACT

Developing a robust and generalizable state representation is essential for overcoming the challenges posed by reinforcement learning tasks that rely on images as input. Recent developments in metric learning, including techniques like deep bisimulation metric approaches, have facilitated the transformation of states into structured representation spaces, allowing the measurement of distances based on task-relevant features. However, these approaches face challenges in handling demanding generalization tasks and scenarios characterized by sparse rewards. Their limited one-step update strategy often falls short of capturing adequate long-term behaviors within their representations. To address these challenges, we present the State Chrono Representation (SCR) approach, which enhances state representations by integrating long-term information alongside the bisimulation metric. SCR learns state distances and measurements within a temporal framework, considering future dynamics and accumulated rewards across current and long-term future states. The resulting representation space not only captures sequential behavioral information but also integrates distances and measurements from the present to the future. This temporal-aware learning strategy does not introduce a significant number of additional parameters for modeling dynamics, ensuring the efficiency of the entire learning process. Comprehensive experiments conducted within DeepMind Control environments reveal that SCR achieves state-of-the-art performance in demanding generalization tasks and scenarios characterized by sparse rewards.

## 1 INTRODUCTION

In deep reinforcement learning (Deep RL), deriving an optimal policy from highly dimensional environmental observations, particularly images, is a critical challenge (Castro, 2020; Gelada et al., 2019; Seo et al., 2022). An RL agent continually receives images that display temporal relationships and substantial spatial redundancy. Redundant and potentially distracting visual inputs make it difficult for the agent to formulate optimal policies. Numerous studies have emphasized the importance of crafting state representations capable of discerning task-relevant information amidst task-irrelevant surroundings. Such representations hold the potential to greatly facilitate the RL process and enhance the generalizability of the learned policies. Consequently, representation learning has been recognized as a cornerstone in the advancement of Deep RL algorithms, garnering increased attention within the RL community (Kirk et al., 2023).

The primary focus of representation learning in reinforcement learning (RL) lies in developing a mapping function to transform high-dimensional observations into low-dimensional embeddings, which mitigates the influence of irrelevant signals to simplify the process of policy learning. Previous research in this area has employed autoencoder-like reconstruction losses (Yarats et al., 2021c; Higgins et al., 2017), yielding impressive outcomes across various visual RL tasks. However, these approaches do not fully account for noise in visual features, which can be vital for accurately reconstructing input images. Data augmentation (Yarats et al., 2021b; Laskin et al., 2020a;b) methods have shown promise in tasks involving noisy observations, primarily enhancing perception models without directly impacting the policy within the context of the Markov Decision Process (MDP). The methods by learning auxiliary tasks (Seo et al., 2022) aim to predict additional tasks related to the environments using the learned representation as input. Nonetheless, these auxiliary tasks are often designed independently of the primary RL objective, potentially limiting their effectiveness.

In recent advancements, behavioral metrics (Ferns et al., 2004; Ferns & Precup, 2014), such as the bisimulation metric (Castro, 2020; Zhang et al., 2021) and MICo (Castro et al., 2021), have emerged to quantify the dissimilarity between two states by considering differences in immediate reward signals and the divergence of next-state distributions. These metric learning methods establish approximate metrics within the representation space, preserving the behavioral similarities among states. State representations are constrained within a structured metric space, wherein each state is positioned or clustered relative to others based on their behavioral distances. Moreover, behavioral metrics have been proven to set an upper bound on state-value discrepancies between corresponding states. By learning behavioral metrics within representations, these methods selectively retain task-relevant features essential for achieving the final RL goal, which involves maximizing the value function and shaping agent behaviors. Conversely, they filter out noise unrelated to state values and behavioral metrics. Behavioral metric approaches have demonstrated remarkable performance in various RL tasks, including control tasks from images, particularly in the presence of noisy images.

However, behavioral metrics encounter challenges in handling demanding generalizable RL tasks and scenarios with sparse reward (Kemertas & Aumentado-Armstrong, 2021). While behavioral metrics can somehow capture long-term behavioral metrics by temporal-difference update mechanism, their reliance on one-step transition data, where the information is limited in the case of sparse reward, significantly hampers learning efficiency. As a result, the representations learned using behavioral metrics may suffer from encoding with non-informative signals, such as sparse rewards (Kemertas & Aumentado-Armstrong, 2021). Some model-based approaches attempt to mitigate these issues by learning transition models, but the task of learning a large transition model with long trajectories places increased demands on computational resources and parameters.

To address the aforementioned challenges, we introduce the State Chrono Representation (SCR) framework, a metric-based approach to learning long-term behavioral representation and accumulated rewards spanning from present to future states. Within the SCR framework, we advocate for the training of two distinct state encoders. An encoder specializes in crafting a state representation for individual states, while the other focuses on generating a *Chronological Embedding*, which encapsulates the relationship between a state and one of its future states. In addition to learning the conventional behavioral metric for state representations, we introduce a novel behavioral metric tailored to temporal state pairs. This new metric is approximated within the chronological embedding space. We also propose an alternative distance metric, distinct from the typical $L_p$ norm, to efficiently approximate this behavioral metric in a lower-dimensional vector space. To infuse long-term rewards information into these representations, we present a "measurement" that quantifies the sum of rewards between the current and future states. Instead of directly regressing this measurement, we impose two constraints on it to restrict its range and value. Note that SCR is a versatile representation learning methodology that can be integrated into any existing RL algorithm.

In summary, our contributions are threefold: 1) We introduce the SCR framework for representation learning with a focus on behavioral metrics involving temporal state pairs. Additionally, we provide a practical method for approximating these metrics.; 2) We develop a novel measurement specifically tailored for temporal state pairs and propose learning algorithms that incorporate this measurement while enforcing two constraints; 3) Our proposed representation demonstrates enhanced generalization and efficiency in challenging generalization tasks, as exemplified by experiments conducted on the Distracting DeepMind Control Suite (Stone et al., 2021; Tunyasuvunakool et al., 2020).

## 2 PRELIMINARY

**Markov Decision Process:** A Markov Decision Process (MDP) is defined as a tuple $\mathcal{M} = (\mathcal{S}, \mathcal{A}, P, r, \gamma)$, where $\mathcal{S}$ represents the state space, consisting of all possible states, $\mathcal{A}$ indicates the action space, consisting of all possible actions which an agent can take in each state. The term $P$ stands for the state transition probability function. Given a current state $s_t \in \mathcal{S}$ and an action $a_t \in \mathcal{A}$ taken, $P(s_{t+1}|s_t, a_t)$ gives the probability of transitioning to any state $s_{t+1} \in \mathcal{S}$. $r : \mathcal{S} \times \mathcal{A} \to \mathbb{R}$ denotes the reward function, which gives the immediate reward $r(s_t, a_t)$ received for taking an action $a_t$ in a state $s_t$. The discount factor, $\gamma \in [0, 1]$, determines the present value of future rewards.

A policy $\pi : \mathcal{S} \to \mathcal{A}$ is a mapping function that determines the action that an agent will take in each state. The goal in MDP is to determine the optimal policy $\pi^*$ that maximizes the expected discounted cumulated reward, $\pi^* = \arg\max_\pi \mathbb{E}[\sum_t \gamma^t r(s_t, \pi(s_t))]$.

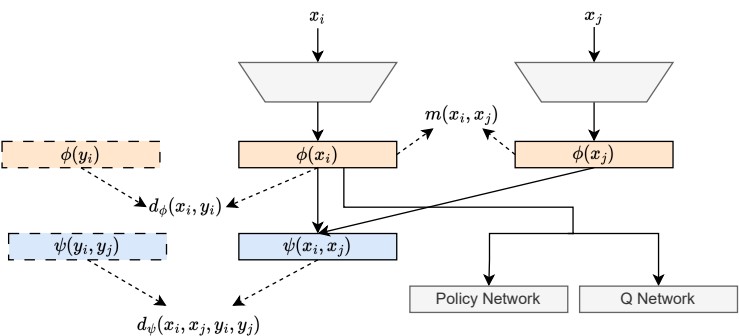

Figure 1: Overall architecture of SCR.

**Behavioral Metric:** The bisimulation metric in DBC (Zhang et al., 2021) defines a pseudometric $d : \mathcal{S} \times \mathcal{S} \to \mathbb{R}$ to measure the distance between two states. A variant of bisimulation metric, known as $\pi$-bisimulation metric, is defined on a given policy $\pi$.

**Theorem 1** ($\pi$-bisimulation metric). *The $\pi$-bisimulation metric update operator $\mathcal{F}_{bisim} : \mathbb{M} \to \mathbb{M}$ is defined as,*

$$\mathcal{F}_{bisim} d(\mathbf{x}, \mathbf{y}) := |r_{\mathbf{x}}^{\pi} - r_{\mathbf{y}}^{\pi}| + \gamma \mathcal{W}(d)(P_{\mathbf{x}}^{\pi}, P_{\mathbf{y}}^{\pi}),$$

*where $\mathbb{M}$ is the space of $d$, $r_{\mathbf{x}}^{\pi} = \sum_{a \in \mathcal{A}} \pi(a|\mathbf{x}) r_{\mathbf{x}}^{a}$, $P_{\mathbf{x}}^{\pi} = \sum_{a \in \mathcal{A}} \pi(a|\mathbf{x}) P_{\mathbf{x}}^{a}$, and $\mathcal{W}$ is the Wasserstein distance. $\mathcal{F}_{bisim}$ has a unique least fixed point $d_{bisim}^{\pi}$.*

MICo (Castro et al., 2021) defines another metric based on sampling the next states without measuring the intractable Wasserstein distance.

**Theorem 2** (MICo distance). *The MICo distance update operator $\mathcal{F}_{MICo} : \mathbb{M} \to \mathbb{M}$ is defined as,*

$$\mathcal{F}_{MICo} d(\mathbf{x}, \mathbf{y}) := |r_{\mathbf{x}}^{\pi} - r_{\mathbf{y}}^{\pi}| + \gamma \mathbb{E}_{\mathbf{x}' \sim P_{\mathbf{x}}^{\pi}, \mathbf{y}' \sim P_{\mathbf{y}}^{\pi}} d(\mathbf{x}', \mathbf{y}'),$$

*$\mathcal{F}_{MICo}$ has a fixed point $d_{MICo}^{\pi}$.*

## 3 STATE CHRONO REPRESENTATION

Despite their capabilities, the bisimulation metric (Zhang et al., 2021) and MICo (Castro et al., 2021) fall short when encoding future information. This limitation can impede the effectiveness of state representations in policy learning. To overcome this shortcoming and integrate future details, we present **State Chrono Representation** (SCR). Figure 1 shows the detailed architecture of SCR.

SCR encompasses two representations: a **state representation** $\phi(\mathbf{x}) \in \mathbb{R}^n$ for a state $\mathbf{x}$ and a **chronological embedding** $\psi(\mathbf{x}_i, \mathbf{x}_j) \in \mathbb{R}^n$ for a state $\mathbf{x}_i$ and its future state $\mathbf{x}_j$. The state representation, $\phi(\mathbf{x})$, is developed through a behavioral metric $d$, which discerns the reward and dynamic divergence between two states. In contrast, the chronological embedding, $\psi(\mathbf{x}_i, \mathbf{x}_j)$, fuses these two state representations using deep learning, highlighting the long-term behavioral correlation between the current state $\mathbf{x}_i$ and future state $\mathbf{x}_j$. A "chronological" behavioral metric is proposed to learn and compute the distance between any two chronological embeddings and is further refined through the Bellman operator-like MSE loss. Moreover, $\phi(\mathbf{x})$ employs an innovative **temporal measurement**, $m$, to assess the transition from the current state to a future one, effectively capturing sequential reward data. This devised temporal measurement operates within defined lower and upper constraints, directing the learning trajectory of both the measurement and state representation.

### 3.1 METRIC LEARNING FOR STATE REPRESENTATION

The state representation encoder $\phi$ is trained by approximating a behavioral metric like the MICo distance. In our model, we adopt a MICo-based metric transformation operator, swapping the sampling-based prediction in MICo with latent dynamics-based modeling to determine the divergence between two subsequent states distribution, drawing parallels with the methodology in SimSR (Zang et al., 2022). The metric update operator for latent dynamics, denoted as $\mathcal{F}$, is defined below.

**Theorem 3.** *Let $\hat{d} : \mathbb{R}^n \times \mathbb{R}^n \to \mathbb{R}$ be a metric in the latent state representation space, $d_\phi(\mathbf{x}_i, \mathbf{y}_{i'}) := \hat{d}(\phi(\mathbf{x}_i), \phi(\mathbf{y}_{i'}))$ be a metric in the state domain. The metric update operator $\mathcal{F}$ is defined as,*

$$\mathcal{F}d_\phi(\mathbf{x}_i, \mathbf{y}_{i'}) = |r_{\mathbf{x}_i} - r_{\mathbf{y}_{i'}}| + \gamma \mathbb{E}_{\substack{\phi(\mathbf{x}_i) \sim \hat{P}(\cdot|\phi(\mathbf{x}_{i+1}), a_{\mathbf{x}_i}) \\ \phi(\mathbf{y}_{i'+1}) \sim \hat{P}(\cdot|\phi(\mathbf{y}_{i'+1}), a_{\mathbf{y}_{i'}})}} \hat{d}(\phi(\mathbf{x}_{i+1}), \phi(\mathbf{y}_{i'+1})), \quad (1)$$

*where $\hat{\mathbb{M}}$ is the space of d, with $a_{\mathbf{x}_i}$ and $a_{\mathbf{y}_{i'}}$ being the actions at states $\mathbf{x}_i$ and $\mathbf{y}_{i'}$, respectively, and $\hat{P}$ is the learned latent dynamics model. $\mathcal{F}$ has a fixed point $d_\phi^\pi$.*

To learn the approximation for $d_\phi^\pi$ in the representation space, the form of distance $\hat{d}$ for low-dimensional vectors must be specified. Castro et al. (2021) demonstrated that a behavioral metric with a sample-based next state distribution divergence is a diffuse metric due to the divergence of the next state distribution being the Łukaszyk-Karmowski distance.

**Definition 3.1** (diffuse metric (Castro et al., 2021)). A function $d : \mathcal{X} \times \mathcal{X} \to \mathbb{R}$ based on the set $\mathcal{X}$ is a diffuse metric if the following axioms hold:
1) $d(\mathbf{a}, \mathbf{b}) \geq 0$ for any $\mathbf{a}, \mathbf{b} \in \mathcal{X}$.
2) $d(\mathbf{a}, \mathbf{b}) = d(\mathbf{b}, \mathbf{a})$ for any $\mathbf{a}, \mathbf{b} \in \mathcal{X}$.
3) $d(\mathbf{a}, \mathbf{b}) + d(\mathbf{b}, \mathbf{c}) \geq d(\mathbf{a}, \mathbf{c})$ for any $\mathbf{a}, \mathbf{b}, \mathbf{c} \in \mathcal{X}$.

MICo offers an approximation of the behavioral metric through an angular distance: $\hat{d}_{MICo}(\mathbf{a}, \mathbf{b}) = \frac{\|\mathbf{a}\|_2^2 + \|\mathbf{b}\|_2^2}{2} + \beta\theta(\mathbf{a}, \mathbf{b})$, where $\mathbf{a}, \mathbf{b} \in \mathbb{R}^n$, $\theta(\mathbf{a}, \mathbf{b})$ represents the angle between vectors $\mathbf{a}$ and $\mathbf{b}$, and $\beta$ is a hyperparameter pre-determined to be 0.1. This distance calculation features a non-zero self-distance, rendering it compatible with expressing the Łukaszyk-Karmowski distance. However, the angle function $\theta(\mathbf{a}, \mathbf{b})$ exclusively considers the angle between $\mathbf{a}$ and $\mathbf{b}$, necessitating computations involving the cosine similarity and arccos function, which can lead to numerical discrepancies. DBC recommends employing the $L_1$ norm with zero self-distance, suitable exclusively for the Wasserstein distance. Meanwhile, SimSR utilizes the cosine distance, derived from the cosine similarity, albeit without fulfilling the triangle inequality and the non-zero self-distance.

To mitigate the aforementioned challenges, we propose a revised distance, $\hat{d}(\mathbf{a}, \mathbf{b})$, in the embedding space, characterized as a diffuse metric. This is mathematically formulated as:

**Definition 3.2.** Define $\hat{d} : \mathbb{R}^n \times \mathbb{R}^n \to \mathbb{R}$ as a distance function, where $\hat{d}(\mathbf{a}, \mathbf{b}) = \sqrt{\|\mathbf{a}\|_2^2 + \|\mathbf{b}\|_2^2 - \mathbf{a}^\top \mathbf{b}}$, for any $\mathbf{a}, \mathbf{b} \in \mathbb{R}^n$.

**Theorem 4.** *$\hat{d}$ is a diffuse metric.*

*Proof.* Refer to Appendix for proofs of Theorem 4. $\qquad\qquad\qquad\qquad\qquad\qquad\qquad\qquad\square$

**Lemma 1** (Non-zero self-distance). *The self-distance of $\hat{d}$ is not stricter to zero, i.e., $\hat{d}(\mathbf{a}, \mathbf{a}) = \|\mathbf{a}\|_2 \geq 0$. This becomes zero if and only if every element in vector $\mathbf{a}$ is zero.*

Theorem 4 validates that $\hat{d}$ is a diffuse metric satisfying triangle inequality, and Lemma 1 shows $\hat{d}$ has non-zero self-distance capable of approximating dynamic divergence that is a Łukaszyk-Karmowski distance. Moreover, The structure of $\hat{d}$ resembles the $L_2$ norm with the exception that the weight before $\mathbf{a}^\top \mathbf{b}$ is -1 instead of -2. Its construction, which only includes vector inner product and square root computations exempting divisions and trigonometric functions, prevents numerical computational issues and simplifies implementation.

To learn the representation function $\phi$, a prevalent approach is minimizing the MSE loss between both ends of Equation 13. The loss, combined with $\hat{d}$ and relative to $\phi$, is expressed as:

$$\mathcal{L}_\phi(\phi) = \mathbb{E}_{\substack{\mathbf{x}_i, \mathbf{y}_{i'}, r_{\mathbf{x}_i}, r_{\mathbf{y}_{i'}} \sim \mathcal{D} \\ \phi(\mathbf{x}_{i+1}), \phi(\mathbf{y}_{i'+1}) \sim \hat{P}}} \left| \hat{d}(\phi(\mathbf{x}_i), \phi(\mathbf{y}_{i'})) - |r_{\mathbf{x}_i} - r_{\mathbf{y}_{i'}}| - \gamma\hat{d}(\phi(\mathbf{x}_{i+1}), \phi(\mathbf{y}_{i'+1})) \right|^2, \quad (2)$$

where $\mathcal{D}$ represents the replay buffer or the sampled rollout for the RL learning process.

### 3.1.1 THE LACK OF LONG-TERM TEMPORAL INFORMATION

The loss $L_\phi(\phi)$ in Equation 2 leans heavily on temporal-difference update mechanism with one-step transitions information. Consequently, it lacks the capacity and efficiency to grasp and encode rich long-term information from trajectories. Encoding temporal information within the representation, while ensuring it remains structured and adheres to behavioral metrics, poses an intricate challenge. To surmount this challenge, we propose two distinct methods, Chronological Embedding (discussed in Section 3.2) and Temporal Measurement (discussed in Section 3.3). Each technique is tailored to harness the temporal essence of a rollout, denoted as $\tau(\mathbf{x}_i; \mathbf{x}_j)$, which represents a sequence that originates from state $\mathbf{x}_i$ and reach its future state $\mathbf{x}_j$.

As illustrated in Figure 2, the chronological embedding seeks to craft an innovative paired state embedding, symbolized as $\psi(\mathbf{x}_i, \mathbf{x}_j)$. This is achieved by transforming state representations $\phi(\mathbf{x}_i)$ and $\phi(\mathbf{x}_j)$ with deep network. The objective for learning $\psi$ is to learn a novel "chronological" behavioral metric, one that measures the distance between rollouts $\tau(\mathbf{x}_i; \mathbf{x}_j)$ and $\tau(\mathbf{y}_{i'}; \mathbf{y}_{j'})$. On the other hand, temporal measurement aspires to compute a special "distance" between states $\mathbf{x}_i$ and $\mathbf{x}_j$. This measurement offers insights into the cumulative rewards amassed throughout the rollout $\tau(\mathbf{x}_i; \mathbf{x}_j)$. However, the learning temporal distance is formidable, and neither single method can claim full mastery over it. Consequently, we synergize both techniques to fortify and elevate the quality of the state representation.

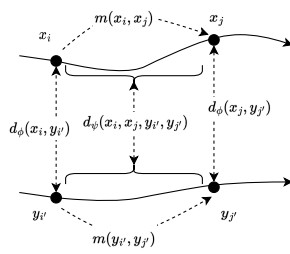

Figure 2: Illustration of an example with two rollout.

## 3.2 CHRONOLOGICAL EMBEDDING

The **chronological embedding**, denoted as $\psi(\mathbf{x}_i, \mathbf{x}_j) \in \mathbb{R}^n$, is tailored to capture the relationship between a given state $\mathbf{x}_i$ and its long-term future states $\mathbf{x}_j$. It is premised on the assumption that both states, $\mathbf{x}_i$ and $\mathbf{x}_j$, originate from the same trajectory. With a focus on capturing extended behavioral knowledge, we introduce a distance function $d_\psi : (\mathcal{S} \times \mathcal{S}) \times (\mathcal{S} \times \mathcal{S}) \to \mathbb{R}$, which is envisaged to mirror the behavioral metric, and allows the encoder $\psi$ to integrate the behavioral information.

Building upon the MICo distance in Theorem 2, we specify the metric update operator $\mathcal{F}_{Chrono}$ for $d_\psi$.

**Theorem 5.** *Let $\mathbb{M}_\psi$ be the space of $d_\psi$. The metric update operator $\mathcal{F}_{Chrono} : \mathbb{M}_\psi \to \mathbb{M}_\psi$ is defined as,*

$$\mathcal{F}_{Chrono}d_\psi(\mathbf{x}_i, \mathbf{x}_j, \mathbf{y}_{i'}, \mathbf{y}_{j'}) = |r_{\mathbf{x}_i} - r_{\mathbf{y}_{i'}}| + \gamma \mathbb{E}_{\mathbf{x}_{i+1} \sim P_\mathbf{x}^\pi, \mathbf{y}_{i'+1} \sim P_\mathbf{y}^\pi} d_\psi(\mathbf{x}_{i+1}, \mathbf{x}_j, \mathbf{y}_{i'+1}, \mathbf{y}_{j'}). \quad (3)$$

*$\mathcal{F}_{Chrono}$ has a fixed point $d_\psi^\pi$.*

Here $d_\psi^\pi$ denotes the "chronological" behavioral metric. Our goal is to closely approximate $d_\psi^\pi$. It quantifies the distance between two sets of states, $(\mathbf{x}_i, \mathbf{x}_j)$ and $(\mathbf{y}_{i'}, \mathbf{y}_{j'})$, considering the immediate reward difference and dynamics divergence. In our strategy to co-learn the encoder $\psi$ with $d_\psi^\pi$, we represent $d_\psi^\pi$ in terms of $\hat{d}$ as $d_\psi^\pi(\mathbf{x}_i, \mathbf{x}_j, \mathbf{y}_{i'}, \mathbf{y}_{j'}) := \hat{d}(\psi((\mathbf{x}_i, \mathbf{x}_j)), \psi(\mathbf{y}_{i'}, \mathbf{y}_{j'}))$, where $\hat{d}$ is defined in Definition 3.2. Similar to Equation 13, we construct $d_\psi^\pi$ to compute the distance in the embedding space.

$$d_\psi^\pi(\mathbf{x}_i, \mathbf{x}_j, \mathbf{y}_{i'}, \mathbf{y}_{j'}) = |r_{\mathbf{x}_i} - r_{\mathbf{y}_{i'}}| + \gamma \mathbb{E}_{\mathbf{x}_{i+1} \sim P_\mathbf{x}^\pi, \mathbf{y}_{i'+1} \sim P_\mathbf{y}^\pi} \hat{d}(\psi(\mathbf{x}_{i+1}, \mathbf{x}_j), \psi(\mathbf{y}_{i'+1}, \mathbf{y}_{j'})). \quad (4)$$

For computational efficiency, the parameters between encoders $\phi$ and $\psi$ are shared. The encoder $\psi$ extracts outputs from $\phi$, and the distance measure is suitably adapted as $d_\psi^\pi(\mathbf{x}_i, \mathbf{x}_j, \mathbf{y}_{i'}, \mathbf{y}_{j'}) := \hat{d}(\psi((\phi(\mathbf{x}_i), \phi(\mathbf{x}_j))), \psi(\phi(\mathbf{y}_{i'}), \phi(\mathbf{y}_{j'})))$. The objective for learning the chronological embedding is formulated as minimizing the MSE loss between both sides of Equation 4 w.r.t $\phi$ and $\psi$,

$$\mathcal{L}_\psi(\psi, \phi) = \mathbb{E}_{\substack{\mathbf{x}_i, \mathbf{x}_j, \mathbf{y}_{i'}, \mathbf{y}_{j'}, r_{\mathbf{x}_i}, \\ r_{\mathbf{y}_{i'}}, \mathbf{x}_{i+1}, \mathbf{y}_{i'+1} \sim \mathcal{D}}} \Big| \hat{d}(\psi((\phi(\mathbf{x}_i), \phi(\mathbf{x}_j))), \psi(\phi(\mathbf{y}_{i'}), \phi(\mathbf{y}_{j'})))$$
$$- |r_{\mathbf{x}_i} - r_{\mathbf{y}_{i'}}| - \gamma \hat{d}(\psi((\phi(\mathbf{x}_{i+1}), \phi(\mathbf{x}_j))), \psi(\phi(\mathbf{y}_{i'+1}), \phi(\mathbf{y}_{j'}))) \Big|^2. \quad (5)$$

The goal of this objective is to nudge embeddings of analogous state sequences closer in the embedded space, bolstering the categorization of congruent behaviors.

### 3.3 Temporal Measurement

To facilitate SCR in acquiring future insights, we introduce **temporal measurement**, a conceptual "distance" to quantify the discrepancies between the current state $\mathbf{x}_i$ and future state $\mathbf{x}_j$. This measurement, $m(\mathbf{x}_i, \mathbf{x}_j)$, aims to measure the differences in state value or the sum of rewards received within the current and future states. We build the approximated measurement upon the state representation $\phi$, i.e., $\hat{m}_\phi(\mathbf{x}_i, \mathbf{x}_j) := \hat{m}(\phi(\mathbf{x}_i), \phi(\mathbf{x}_j))$. where $\hat{m} : \mathbb{R}^n \times \mathbb{R}^n \to \mathbb{R}$ is a non-parametric asymmetrical metric function further detailed in Section 3.3.1. This approach organizes the representation space of $\phi(\mathbf{x})$ around the "distance" $m$, enabling the structured representation of $\phi(\mathbf{x})$ to hold sufficient information to plan for future states.

We propose $m$ to represent the expected discounted accumulated rewards, obtained by an optimal policy $\pi^*$ from state $\mathbf{x}_i$ to state $\mathbf{x}_j$:

$$m(\mathbf{x}_i, \mathbf{x}_j) = \mathbb{E}_{\pi^*}\left[\sum_{t=0}^{j-i} \gamma^t r_{\mathbf{s}_t}\Bigg| \mathbf{s}_0 = \mathbf{x}_i, \mathbf{s}_{j-i} = \mathbf{x}_j\right]. \tag{6}$$

However, it is non-trivial to obtain $m(\mathbf{x}_i, \mathbf{x}_j)$ because the optimal policy $\pi^*$ is unknown and is, de facto, the primary goal of the RL task. Instead of directly approximating $m(\mathbf{x}_i, \mathbf{x}_j)$, we learn the approximation $\hat{m}(\mathbf{x}_i, \mathbf{x}_j)$ in an alternative way that ensures it is located in a feasible range covering the true $m(\mathbf{x}_i, \mathbf{x}_j)$. To construct this range, we introduce two constraints.

The first constraint, which is considered as a **lower** boundary, asserts that the expected discounted cumulative reward collected by any policy $\pi$, optimal or otherwise, cannot surpass $m$:

$$\mathbb{E}_{\pi}\left[\sum_{t=0}^{j-i} \gamma^t r_{\mathbf{s}_t}\Bigg| \mathbf{s}_0 = \mathbf{x}_i, \mathbf{s}_{j-i} = \mathbf{x}_j\right] \leq m(\mathbf{x}_i, \mathbf{x}_j). \tag{7}$$

This constraint is constructed based on an assumption that any sub-optimal policy is inferior to the optimal policy. Based on the constraint in Equation 7, we propose the first objective for learning the approximation $\hat{m}_\phi$:

$$\mathcal{L}_{low}(\phi) = \mathbb{E}_{\tau(\mathbf{x}_i;\mathbf{x}_j)\sim\pi}\left|ReLU\left(\sum_{t=0}^{j-i} \gamma^t r_{\mathbf{x}_t} - \hat{m}(\phi(\mathbf{x}_i), \phi(\mathbf{x}_j))\right)\right|^2, \tag{8}$$

where $ReLU(x) = x^+ = max(0, x)$. This objective becomes non-zero when the constraint in Equation 7 is not satisfied, pushing the value of $m(\phi(\mathbf{x}_i), \phi(\mathbf{x}_j))$ to be larger until it becomes larger than the sampled reward sum.

The second constraint, corresponding to **upper** boundary, is proposed according to inspiration from the triangle inequality. The absolute value $|m(\mathbf{x}_i, \mathbf{x}_j)|$ is limited by the following inequality,

$$|m(\mathbf{x}_i, \mathbf{x}_j)| \leq d(\mathbf{x}_i, \mathbf{y}_{i'}) + |m(\mathbf{y}_{i'}, \mathbf{y}_{j'})| + d(\mathbf{x}_j, \mathbf{y}_{j'}), \tag{9}$$

where $d$ is the behavioral metric introduced in Section 3.1. The right-hand side represents the longer path from $\mathbf{x}_i$ to $\mathbf{x}_j$. This inequality demonstrates that the absolute temporal measurement $|m(\mathbf{x}_i, \mathbf{x}_j)|$ is no greater than the sum of behavioral metrics at the beginning states ($\mathbf{x}_i$ and $\mathbf{y}_{i'}$), i.e. $d(\mathbf{x}_i, \mathbf{y}_{i'})$, and end states pair ($\mathbf{x}_j$ and $\mathbf{y}_{j'}$), i.e. $d(\mathbf{x}_j, \mathbf{y}_{j'})$, respectively, plus the measurement $|m(\mathbf{y}_i, \mathbf{y}_j)|$. This constraint leads to the following formulation of the 2nd objective for training $\hat{m}_\phi$:

$$\begin{aligned}\mathcal{L}_{up}(\phi) = \Bigg|ReLU\Big(&|\hat{m}(\phi(\mathbf{x}_i), \phi(\mathbf{x}_j))|\\ &- \text{stop\_grad}\left(\hat{d}((\phi(\mathbf{x}_i), \phi(\mathbf{y}_{i'}))) + \hat{d}((\phi(\mathbf{x}_j), \phi(\mathbf{y}_{j'}))) + m(\phi(\mathbf{y}_{i'}), \phi(\mathbf{y}_{j'}))\right)\Big)\Bigg|^2\end{aligned} \tag{10}$$

This objective pushes down the value of $\hat{m}_\phi$ when the constraint in Equation 9 is not satisfied.

By optimizing both constraints in a unified manner, we ensure the approximated temporal measurement, $m$, is anchored within a specific range, bounded by the lower and upper constraints. The overall objective for $\hat{m}$ is formulated as:

$$\mathcal{L}_{\hat{m}}(\phi) = \mathcal{L}_{low}(\phi) + \mathcal{L}_{up}(\phi). \tag{11}$$

### 3.3.1 Asymmetrical Metric Function for $\hat{m}$

The measurement $\hat{m}(\mathbf{x}_i, \mathbf{x}_j)$ designed to measure the distance regarding the rewards should be **asymmetrical** with respect to $\mathbf{x}_i$ and $\mathbf{x}_j$. This is predicated on our assumption that state $\mathbf{x}_i$ precedes $\mathbf{x}_j$, making its relation distinctly different from the progression from $\mathbf{x}_j$ to $\mathbf{x}_i$. Recently research studies the quasimetric in deep learning (Pitis et al., 2020; Wang & Isola, 2022b;a) and develop various methodologies to compute asymmetrical distances. In our method, we opt to leverage Interval Quasimetric Embedding (IQE) (Wang & Isola, 2022a) to implement $\hat{m}$.

### 3.4 Overall Objective

As shown in Figure 1, the encoders are designed to predict state representations $\phi(\mathbf{x})$ for individual states and chrono embedding $\psi(\mathbf{x}_i, \mathbf{x}_j)$ for capturing the relationship existing between that states $\mathbf{x}_i$ and $\mathbf{x}_j$. The measurement $\hat{m}$ is subsequently computed based on $\phi(\mathbf{x}_i)$ and $\phi(\mathbf{x}_j)$ to account for the accumulated rewards in between these states. The components $\psi$ and $\hat{m}$ collaboratively enhance the state representations $\phi$ to capture the temporal information and its predictive capabilities for future insight. We will show the necessity of $\psi$ and $\hat{m}$ in the ablation study which is detailed in Section 4.3. Therefore, a comprehensive objective is formulated in a unified manner:

$$\mathcal{L}(\phi, \psi) = \mathcal{L}_\phi(\phi) + \mathcal{L}_\psi(\psi, \phi) + \mathcal{L}_{\hat{m}}(\phi). \tag{12}$$

Our method, denoted as SCR, possesses the flexibility to be integrated with a broad spectrum of deep RL algorithms. These algorithms can effectively utilize the representation $\phi(\mathbf{x})$ as an integral input component. In our implementation, we employ Soft Actor-Critic (SAC) (Haarnoja et al., 2018) as our foundational RL algorithm. The state representation serves as the input state for the policy network and Q-value network in SAC. Other implementation details are referred to Appendix A.2.

## 4 Experiments

**Benchmarks.** The primary objective of our proposed SCR is to cultivate a versatile and generalizable representation for deep RL when dealing with high-dimensional observation. To assess its efficiency, we conduct experiments utilizing the DeepMind Control Suite (DM_Control) environment with rendered pixels observations (Tunyasuvunakool et al., 2020) and a distraction setting, Distracting Control Suite (Stone et al., 2021). This environment utilizes the MuJoCo physics engine, providing pixel observations for a set of continuous control tasks. It includes rich testing scenarios, especially given its inclusion of background distractions and camera pose distractions, which simulate real-world complexities with camera inputs. Specifically,

- **Default setting.** We evaluate our SCR on seven tasks in DM_Control compared with other RL approaches. Each frame is rendered $3 \times 84 \times 84$ pixels, as demonstrated in Figure 3. We stack three frames as states and feed them into the RL agents.
- **Distraction setting.** To evaluate to generalizability of SCR, we perform experiments on DDM_Control with distraction. The distraction (as shown in Figure 4) includes 1) **background video** distraction, replacing the clean and simple background with a natural video; 2) **object color** distraction, slightly changing the color of the bodies of the robot; and 3) **camera pose** distraction, randomizing the camera pose of position and angle for rendering from the simulator. We observe that tasks become very hard if camera pose distraction is applied.

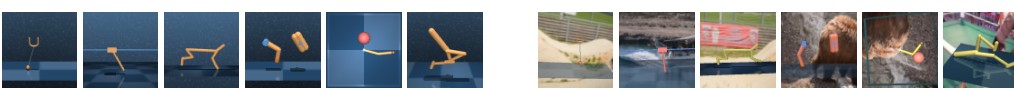

Figure 3: Examples of observations from DM_Control with default setting.

Figure 4: Examples of observations from DM_Control with distraction setting.

**Baselines.** For a comprehensive understanding of our method's performance, we benchmark it against prominent algorithms in the domain, including: 1) SAC (Haarnoja et al., 2018), a baseline deep RL method for continuous control; 2) DrQ (Yarats et al., 2021b), a data augmentation method using random crop; 3) DBC (Zhang et al., 2021), representation learning with the bisimulation metric; and 4) SimSR (Zang et al., 2022), representation learning with the behavioral metric approximated by the cosine distance.

|  | SAC | DrQ | DBC | SimSR | SCR |
|---|---|---|---|---|---|
| ball_in_ cup-catch | 450.6±452.0 | **964.2**±20.3 | 125.2±309.3 | **961.2**±21.3 | **962.8**±29.4 |
| cartpole-swing_up | 808.0±54.6 | 824.8±40.3 | 346.7±79.2 | **866.9**±5.3 | 858.6±4.3 |
| cartpole-swing_up_sparse | 12.5±9.5 | 762.2±32.4 | 220.2±218.0 | 725.3±152.8 | **828.0**±13.3 |
| cheetah-run | 366.0±72.0 | 491.2±30.4 | 372.6±20.9 | **809.7**±29.3 | 734.4±26.8 |
| finger-spin | 436.3±10.7 | **958.2**±11.4 | 413.8±7.8 | **973.0**±11.3 | 968.3±10.2 |
| reacher-easy | 381.4±420.9 | **977.2**±13.6 | 222.3±354.7 | 83.5±166.6 | 865.5±209.5 |
| walker-walk | 313.3±100.1 | 914.8±45.8 | 384.7±167.0 | 934.7±41.9 | **937.1**±35.1 |

Table 1: Result scores on DeepMind Control Suite with the default setting at 500K steps. Each result is written in the format of mean±std.

|  | SAC | DrQ | DBC | SimSR | SCR |
|---|---|---|---|---|---|
| ball_in_ cup-catch | 38.4±145.4 | **257.9**±398.3 | 24.6±153.9 | 145.0±345.2 | 171.5±251.4 |
| cartpole-swing_up | 220.6±48.3 | 276.1±114.9 | 108.6±46.1 | 101.8±63.1 | **494.4**±90.6 |
| cartpole-swing_up_sparse | 2.8±5.8 | 0.6±1.7 | 0.0±0.0 | 0.0±0.0 | **30.7**±6.5 |
| cheetah-run | 154.3±63.0 | 161.2±64.1 | 10.1±2.2 | 10.7±2.3 | **310.0**±115.9 |
| finger-spin | 88.7±94.9 | 628.1±279.7 | 1.0±3.1 | 0.3±0.9 | **851.4**±28.8 |
| reacher-easy | 89.8±180.5 | 115.5±252.8 | 179.5±301.4 | 93.4±123.1 | **209.8**±323.5 |
| walker-walk | 170.1±47.5 | 28.8±11.0 | 26.0±8.7 | 28.2±12.3 | **543.1**±75.0 |

Table 2: Result scores on DM_Control with distraction setting at 500K step. Distraction includes background, robot body color, and camera pose. Each result is written in the format of mean±std.

## 4.1 RESULTS ON THE DEFAULT SETTING

In order to verify the sample efficiency of our method, we compare it with other methods on seven tasks in DM_Control: ball_in_ cup-catch, cartpole-swing_up, cartpole-swing_up_sparse, cheetah-run, finger-spin, reacher-easy and walker-walk. We train each method in each task for 500K steps. Table 1 reports the scores evaluated at the end of training. All experimental results are averaged over 3 runs. We can observe that SCR has comparable results with the augmentation method DrQ and the state-of-the-art behavioral metric approach SimSR. Given that the maximum achievable returns for a DM_Control task stands at 1000, a policy that collects scores around 900 is nearly optimal. These outcomes underscore the potency of SCR in mastering standard RL control tasks.

## 4.2 RESULTS ON THE DISTRACTION SETTING

To further evaluate the generalization ability of our method, we perform comparison experiments on DM_Control with Distraction. We use the same training configuration as with the default setting. The camera-pose distraction presents a challenge for metric-centric methods like DBC and SimSR, primarily due to the significant distortion of the robot shape and position in the image state. Table 2 shows the results scores. Our method outperforms all other methods, including sparse reward task cartpole-swing_up_sparse, where other methods receive almost zero scores. DrQ outperforms other behavioral metric methods as its random cropping facilitates better alignment of robot position.

## 4.3 ABLATION STUDY

To evaluate the impact of each component in the proposed SCR, we perform an ablation study where certain components were selectively removed or substituted. Figure 5 shows the training curves on cheetah-run and walker-walk under distracting setting. **SCR** is the full model of the proposed method. **SCR w/o** $\psi$ removes the chronological embedding

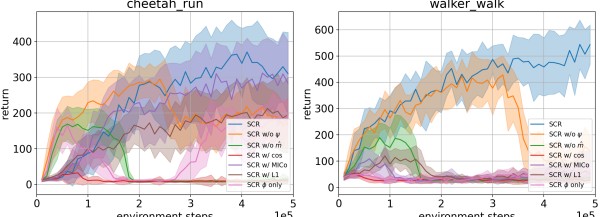

Figure 5: Ablation study on cheetah-run (left) and walker-walk (right) in distracting setting. Curves are evaluation scores average on 3 runs and shadow shapes are std.

$\psi$. **SCR w/o** $\hat{m}$ refers to exclusion of the approximation $\hat{m}$. **SCR w/ cos** replaces the distance function $\hat{d}$ for computing metrics on representation space with cosine distance, akin to SimSR does.

**SCR w/ MICo** replaces $\hat{d}$ with MICo's angular distance. **SCR w/ L1** replaces $\hat{d}$ with L1 distance as adopted by DBC. **SCR w/ $\phi$ only** removes losses $L_\psi(\phi, \psi)$ and $L_{\hat{m}}(\phi)$ but keep $L_\phi(\phi)$. The results show the superior performance of the full model and the importance of $\psi$ and $\hat{m}$. The absence of these components can lead to worse performance and unstable training.

## 4.4 RESULTS ON BACKGROUND DISTRACTION ONLY

We provide another version of DM_Control with distraction which includes only background distraction by replacing with greyscale videos. This experiment setup follows DBC and SimSR. Table 3 shows the experiment results. It is crucial to note that tasks limited to only background distractions are considerably simpler compared to those with object color and camera pose distractions.

|  | DBC | SimSR | SCR |
|---|---|---|---|
| ball_in_ cup-catch | 174.7±340.5 | 71.7±230.9 | **688.8**±292.2 |
| cartpole-swing_up | 120.3±57.1 | 840.4±20.8 | **855.6**±5.0 |
| cartpole-swing_up_sparse | 559.0±145.0 | 694.5±276.8 | **792.1**±41.3 |
| cheetah-run | 295.0±72.5 | 540.7±166.6 | **584.9**±61.6 |
| finger-spin | 526.9±69.4 | 952.5±13.5 | **960.4**±7.4 |
| reacher-easy | 133.3±231.5 | 128.8±143.3 | **286.8**±401.1 |
| walker-walk | 222.0±30.3 | **920.4**±76.8 | 880.8±47.0 |

Table 3: Result scores on DeepMind Control Suite with the distraction setting at 500K steps. Distraction includes background video distraction only. Each result is written in the format of mean±std.

We perform these experiments to fairly compare with DBC and SimSR. The results show that SimSR performs quite well under this setting and our SCR achieves comparable results with SimSR.

## 5 RELATED WORK

Recent studies have investigated representation learning in RL in many ways. Previous works (Higgins et al., 2017; Lee et al., 2020a; Yarats et al., 2021c) train autoencoder to encode image states into low-dimensional latent embeddings which improve the visual perception and accelerate policy learning. Approaches (Yarats et al., 2021b; Laskin et al., 2020a;b; Yarats et al., 2021a; Stooke et al., 2021) utilize data augmentation, e.g., random crop or noise injection, accompanied with contrastive loss to learn better generalizable state representations. Auxiliary tasks approaches (Lee et al., 2020b; Seo et al., 2022; Hafner et al., 2019) learn representation by predicting auxiliary tasks to extract more information from environments.

Recent research with metric learning method for RL learns to measure distance on state representations. Some approaches learn to approximate bisimulation metric (Zhang et al., 2021; Kemertas & Aumentado-Armstrong, 2021) while other approaches learn sample-based distance (Zang et al., 2022; Castro et al., 2021). Goal-based RL have harnessed bisimulation metrics for state representation (Hansen-Estruch et al., 2022). Additionally, a recent work introduces quasimetrics learning as a fresh RL objective for cost MDPs (Wang et al., 2023).

## 6 CONCLUSION

The primary challenge of deep RL lies in cultivating an optimal policy from intricate, high-dimensional noisy observations, such as images. In this work, we propose a novel metric-based representation framework State Chrono Representation (SCR) that leverages temporal dynamics in RL. SCR stands out by fusing the foundational principles of behavioral metrics with a holistic appreciation for long-term state dynamics. Our proposed method, while acknowledging past strides made using behavioral metrics, accentuates the need for a long-term vision in state representation, plugging gaps left by the one-step-focused models. Furthermore, by innovatively measuring rewards over temporal trajectories and weaving this measurement through representations, SCR pushes the frontier of efficient and adaptive RL representation. Our comprehensive experiments underscore the potency of this approach, especially in intricate environments laden with distractions. As the field advances, SCR serves as a testament to the blend of temporal relevance and behavioral metrics, hinting at the trajectory future representation learning endeavors might take.

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

## A APPENDIX

### A.1 PROOFS

#### A.1.1 PROOF OF THEOREM 3

**Theorem 3.** Let $\hat{d} : \mathbb{R}^n \times \mathbb{R}^n \to \mathbb{R}$ be a metric in the latent state representation space, $d_\phi(\mathbf{x}_i, \mathbf{y}_{i'}) := \hat{d}(\phi(\mathbf{x}_i), \phi(\mathbf{y}_{i'}))$ be a metric in the state domain. The metric update operator $\mathcal{F}$ is defined as,

$$\mathcal{F}d_\phi(\mathbf{x}_i, \mathbf{y}_{i'}) = |r_{\mathbf{x}_i} - r_{\mathbf{y}_{i'}}| + \gamma \mathbb{E}_{\substack{\phi(\mathbf{x}_i) \sim \hat{P}(\cdot|\phi(\mathbf{x}_{i+1}), a_{\mathbf{x}_i}) \\ \phi(\mathbf{y}_{i'+1}) \sim \hat{P}(\cdot|\phi(\mathbf{y}_{i'+1}), a_{\mathbf{y}_{i'}})}} \hat{d}(\phi(\mathbf{x}_{i+1}), \phi(\mathbf{y}_{i'+1})), \quad (13)$$

where $\hat{\mathbb{M}}$ is the space of $d$, with $a_{\mathbf{x}_i}$ and $a_{\mathbf{y}_{i'}}$ being the actions at states $\mathbf{x}_i$ and $\mathbf{y}_{i'}$, respectively, and $\hat{P}$ is the learned latent dynamics model. $\mathcal{F}$ has a fixed point $d_\phi^\pi$.

*Proof.* We follow the proof techniques from (Castro et al., 2021) and (Zang et al., 2022). By substituting $\hat{d}(\phi(\mathbf{x}_{i+1}), \phi(\mathbf{y}_{i'+1}))$ with $d_\phi(\mathbf{x}_{i+1}, \mathbf{y}_{i'+1})$, we have

$$\mathcal{F}d_\phi(\mathbf{x}_i, \mathbf{y}_{i'}) = |r_{\mathbf{x}_i} - r_{\mathbf{y}_{i'}}| + \gamma \mathbb{E}_{\substack{\mathbf{x}_i \sim \hat{P}(\cdot|\phi(\mathbf{x}_{i+1}), a_{\mathbf{x}_i}) \\ \mathbf{y}_{i'+1} \sim \hat{P}(\cdot|\phi(\mathbf{y}_{i'+1}), a_{\mathbf{y}_{i'}})}} d_\phi(\mathbf{x}_{i+1}, \mathbf{y}_{i'+1}). \quad (14)$$

The operator $\mathcal{F}d_\phi$ is a contraction mapping with respect to the $L_\infty$ norm because,

$$|\mathcal{F}d_\phi(\mathbf{x}, \mathbf{y}) - \mathcal{F}d_{\phi'}(\mathbf{x}, \mathbf{y})| = \left| \gamma \mathbb{E}_{\substack{\mathbf{x}_i \sim \hat{P}(\cdot|\phi(\mathbf{x}_{i+1}), a_{\mathbf{x}_i}) \\ \mathbf{y}_{i'+1} \sim \hat{P}(\cdot|\phi(\mathbf{y}_{i'+1}), a_{\mathbf{y}_{i'}})}} (d_\phi - d_{\phi'})(\mathbf{x}_{i+1}, \mathbf{y}_{i'+1}) \right| \quad (15)$$
$$\leq \gamma \|(d_\phi - d_{\phi'})(\mathbf{x}_{i+1}, \mathbf{y}_{i'+1})\|_\infty.$$

By Banach's fixed point theorem, operator $\mathcal{F}$ has a fixed point $d_\phi^\pi$. □

### A.1.2 PROOF OF THEOREM 4

To prove the distance $\hat{d}$ is a diffuse metric, we need to prove that $\hat{d}$ satisfies all of three axioms in the Definition 3.1 (definition of diffuse metric).

**Lemma 2.** $\|\mathbf{a}\|_2^2 + \|\mathbf{b}\|_2^2 - \mathbf{a}^\top \mathbf{b} \geq 0$ *for any* $\mathbf{a}, \mathbf{b} \in \mathbb{R}$.

*Proof.* Because $\mathbf{a}^\top \mathbf{b} \leq \|\mathbf{a}\|\|\mathbf{b}\|$, we have,

$$
\begin{aligned}
&\|\mathbf{a}\|^2 + \|\mathbf{b}\|^2 - \mathbf{a}^\top \mathbf{b} \\
\geq &\|\mathbf{a}\|^2 + \|\mathbf{b}\|^2 - \|\mathbf{a}\|\|\mathbf{b}\| \\
\geq &\|\mathbf{a}\|^2 + \|\mathbf{b}\|^2 - 2\|\mathbf{a}\|\|\mathbf{b}\| \\
= &(\|\mathbf{a}\| - \|\mathbf{b}\|)^2 \\
\geq &0.
\end{aligned}
\tag{16}
$$

□

This lemma indicates that the term under the square root of $\hat{d}$ is always non-negative. $\hat{d}$ is able to measure any two vectors $\mathbf{a}, \mathbf{b} \in \mathbb{R}^n$.

**Lemma 3** (Non-negative). $\hat{d}(\mathbf{a}, \mathbf{b}) \geq 0$ *for any* $\mathbf{a}, \mathbf{b} \in \mathbb{R}$.

*Proof.* By definition, the square root is non-negative. □

**Lemma 4** (Symmetric). $\hat{d}(\mathbf{a}, \mathbf{b}) = \hat{d}(\mathbf{b}, \mathbf{a})$

*Proof.* $\hat{d}(\mathbf{a}, \mathbf{b}) = \sqrt{\|\mathbf{a}\|_2^2 + \|\mathbf{b}\|_2^2 - \mathbf{a}^\top \mathbf{b}} = \sqrt{\|\mathbf{b}\|_2^2 + \|\mathbf{a}\|_2^2 - \mathbf{b}^\top \mathbf{a}} = \hat{d}(\mathbf{b}, \mathbf{a})$ □

**Lemma 5** (Triangle inequality). $\hat{d}(\mathbf{a}, \mathbf{b}) + \hat{d}(\mathbf{b}, \mathbf{c}) \geq \hat{d}(\mathbf{a}, \mathbf{c})$, *for any* $\mathbf{a}, \mathbf{b}, \mathbf{c} \in \mathbb{R}$.

*Proof.* To prove this lemma, it is equivalent to prove the following inequality by definition of $\hat{d}$,

$$
\sqrt{\|\mathbf{a}\|^2 + \|\mathbf{b}\|^2 - \mathbf{a}^\top \mathbf{b}} + \sqrt{\|\mathbf{b}\|^2 + \|\mathbf{c}\|^2 - \mathbf{b}^\top \mathbf{c}} \geq \sqrt{\|\mathbf{a}\|^2 + \|\mathbf{c}\|^2 - \mathbf{a}^\top \mathbf{c}}.
\tag{17}
$$

Because $-\|\mathbf{x}\|\|\mathbf{y}\| \leq \mathbf{x}^\top \mathbf{y} \leq \|\mathbf{x}\|\|\mathbf{y}\|, \forall \mathbf{x}, \mathbf{y}$, we have

$$
\begin{aligned}
&\sqrt{\|\mathbf{a}\|^2 + \|\mathbf{b}\|^2 - \mathbf{a}^\top \mathbf{b}} + \sqrt{\|\mathbf{b}\|^2 + \|\mathbf{c}\|^2 - \mathbf{b}^\top \mathbf{c}} \\
\geq &\sqrt{\|\mathbf{a}\|^2 + \|\mathbf{b}\|^2 - \|\mathbf{a}\|\|\mathbf{b}\|} + \sqrt{\|\mathbf{b}\|^2 + \|\mathbf{c}\|^2 - \|\mathbf{b}\|\|\mathbf{c}\|},
\end{aligned}
\tag{18}
$$

and

$$
\sqrt{\|\mathbf{a}\|^2 + \|\mathbf{c}\|^2 + \|\mathbf{a}\|\|\mathbf{c}\|} \geq \sqrt{\|\mathbf{a}\|^2 + \|\mathbf{c}\|^2 - \mathbf{a}^\top \mathbf{c}}.
\tag{19}
$$

If

$$
\sqrt{\|\mathbf{a}\|^2 + \|\mathbf{b}\|^2 - \|\mathbf{a}\|\|\mathbf{b}\|} + \sqrt{\|\mathbf{b}\|^2 + \|\mathbf{c}\|^2 - \|\mathbf{b}\|\|\mathbf{c}\|} \geq \sqrt{\|\mathbf{a}\|^2 + \|\mathbf{c}\|^2 + \|\mathbf{a}\|\|\mathbf{c}\|}
\tag{20}
$$

is true, then inequality (17) is true and Lemma 5 is proven. To prove inequality (20), we can take squares on both sides without sign changing because both sides are non-negative. Then we have,

$$
\left( \sqrt{\|\mathbf{a}\|^2 + \|\mathbf{b}\|^2 - \|\mathbf{a}\|\|\mathbf{b}\|} + \sqrt{\|\mathbf{b}\|^2 + \|\mathbf{c}\|^2 - \|\mathbf{b}\|\|\mathbf{c}\|} \right)^2 \geq \|\mathbf{a}\|^2 + \|\mathbf{c}\|^2 + \|\mathbf{a}\|\|\mathbf{c}\|.
\tag{21}
$$

To prove inequality (20), it is equivalent to prove inequality (21). Expand and simplify inequality (21), we have

$$
2\sqrt{\|\mathbf{a}\|^2 + \|\mathbf{b}\|^2 - \|\mathbf{a}\|\|\mathbf{b}\|}\sqrt{\|\mathbf{b}\|^2 + \|\mathbf{c}\|^2 - \|\mathbf{b}\|\|\mathbf{c}\|} \geq -2\|\mathbf{b}\|^2 + \|\mathbf{a}\|\|\mathbf{b}\| + \|\mathbf{b}\|\|\mathbf{c}\| + \|\mathbf{a}\|\|\mathbf{c}\|.
\tag{22}
$$

The left-hand side of inequality (22) is non-negative.

1) if right hand side $-2\|\mathbf{b}\|^2 + \|\mathbf{a}\|\|\mathbf{b}\| + \|\mathbf{b}\|\|\mathbf{c}\| + \|\mathbf{a}\|\|\mathbf{c}\| < 0$, then inequality (22) is proven and backtrace to Lemma 5 is proven.

2) if right hand side $-2\|\mathbf{b}\|^2 + \|\mathbf{a}\|\|\mathbf{b}\| + \|\mathbf{b}\|\|\mathbf{c}\| + \|\mathbf{a}\|\|\mathbf{c}\| \geq 0$, we take square on both sides of inequality (22) and have,

$$4(\|\mathbf{a}\|^2 + \|\mathbf{b}\|^2 - \|\mathbf{a}\|\|\mathbf{b}\|)(\|\mathbf{b}\|^2 + \|\mathbf{c}\|^2 - \|\mathbf{b}\|\|\mathbf{c}\|) \geq (-2\|\mathbf{b}\|^2 + \|\mathbf{a}\|\|\mathbf{b}\| + \|\mathbf{b}\|\|\mathbf{c}\| + \|\mathbf{a}\|\|\mathbf{c}\|)^2. \tag{23}$$

To prove inequality (23), we let left-hand side subtract right-hand side,

$$4(\|\mathbf{a}\|^2 + \|\mathbf{b}\|^2 - \|\mathbf{a}\|\|\mathbf{b}\|)(\|\mathbf{b}\|^2 + \|\mathbf{c}\|^2 - \|\mathbf{b}\|\|\mathbf{c}\|) - (-2\|\mathbf{b}\|^2 + \|\mathbf{a}\|\|\mathbf{b}\| + \|\mathbf{b}\|\|\mathbf{c}\| + \|\mathbf{a}\|\|\mathbf{c}\|)^2$$
$$= 3\|\mathbf{a}\|^2\|\mathbf{b}\|^2 + 3\|\mathbf{a}\|^2\|\mathbf{c}\|^2 + 3\|\mathbf{b}\|^2\|\mathbf{c}\|^2 - 6\|\mathbf{a}\|^2\|\mathbf{b}\|\|\mathbf{c}\| - 6\|\mathbf{a}\|\|\mathbf{b}\|\|\mathbf{c}\|^2 + 6\|\mathbf{a}\|\|\mathbf{b}\|^2\|\mathbf{c}\|$$
$$= 3(\|\mathbf{a}\|\|\mathbf{b}\| + \|\mathbf{b}\|\|\mathbf{c}\| - \|\mathbf{a}\|\|\mathbf{c}\|)^2$$
$$\geq 0. \tag{24}$$

Therefore, inequality (23) is proven. Consequently, inequality (22) in the case of $-2\|\mathbf{b}\|^2 + \|\mathbf{a}\|\|\mathbf{b}\| + \|\mathbf{b}\|\|\mathbf{c}\| + \|\mathbf{a}\|\|\mathbf{c}\| \geq 0$ is proven. Summarize with the case of $-2\|\mathbf{b}\|^2 + \|\mathbf{a}\|\|\mathbf{b}\| + \|\mathbf{b}\|\|\mathbf{c}\| + \|\mathbf{a}\|\|\mathbf{c}\| < 0$, inequality (22) is proven and Lemma 5 is proven.

$\square$

**Theorem 6.** $\hat{d}$ *is a diffuse metric.*

*Proof.* By summarizing Lemma 3, 4, and 5, function $\hat{d}$ holds the three axioms of diffuse metric. Therefore, function $\hat{d}$ is a diffuse metric. $\square$

### A.1.3 PROOF OF THEOREM 5

**Theorem 5.** *Let $\mathbb{M}_\psi$ be the space of $d_\psi$. The metric update operator $\mathcal{F}_{Chrono} : \mathbb{M}_\psi \to \mathbb{M}_\psi$ is defined as,*

$$\mathcal{F}_{Chrono} d_\psi(\mathbf{x}_i, \mathbf{x}_j, \mathbf{y}_{i'}, \mathbf{y}_{j'}) = |r_{\mathbf{x}_i} - r_{\mathbf{y}_{i'}}| + \gamma \mathbb{E}_{\mathbf{x}_{i+1}\sim P_{\mathbf{x}}^\pi, \mathbf{y}_{i'+1}\sim P_{\mathbf{y}}^\pi} d_\psi(\mathbf{x}_{i+1}, \mathbf{x}_j, \mathbf{y}_{i'+1}, \mathbf{y}_{j'}). \tag{25}$$

*$\mathcal{F}_{Chrono}$ has a fixed point.*

*Proof.* We follow the proof in Section A.1.1. $\mathcal{F}_{Chrono}$ is contraction mapping because

$$|\mathcal{F}_{Chrono} d_\psi(\mathbf{x}_i, \mathbf{x}_j, \mathbf{y}_{i'}, \mathbf{y}_{j'}) - \mathcal{F}_{Chrono} d_{\psi'}(\mathbf{x}_i, \mathbf{x}_j, \mathbf{y}_{i'}, \mathbf{y}_{j'})|$$
$$= \left| \gamma \mathbb{E}_{\mathbf{x}_{i+1}\sim P_{\mathbf{x}}^\pi, \mathbf{y}_{i'+1}\sim P_{\mathbf{y}}^\pi}(d_\psi - d_{\psi'})(\mathbf{x}_{i+1}, \mathbf{x}_j, \mathbf{y}_{i'+1}, \mathbf{y}_{j'}) \right| \tag{26}$$
$$\geq \gamma \|(d_\psi - d_{\psi'})(\mathbf{x}_{i+1}, \mathbf{x}_j, \mathbf{y}_{i'+1}, \mathbf{y}_{j'})\|_\infty$$

By Banach's fixed point theorem, operator $\mathcal{F}_{Chrono}$ has a fixed point $d_\psi^\pi$. $\square$

## A.2 IMPLEMENTATION DETAILS

### A.2.1 IMPLEMENTATION OF $\hat{m}$

We adopt IQE (Wang & Isola, 2022a) to implement $\hat{m}$. Given two vectors $\mathbf{a}, \mathbf{b} \in \mathbb{R}^n$, reshaping to $\mathbb{R}^{k \times l}$ where $k \times l = n$, IQE first computes the union of the interval for each component:

$$d_i(\mathbf{a}, \mathbf{b}) = |\bigcup_{j=1}^{l}[\mathbf{a}_{ij}, \max(\mathbf{a}_{ij}, \mathbf{b}_{ij})]|, \forall i = 1, 2, ..., k, \tag{27}$$

where $[\cdot, \cdot]$ is the interval on the real line. Then it computes the distance among all components $d_i$ as

$$d_{IQE}(\mathbf{a}, \mathbf{b}) = \alpha \cdot \max(d_1(\mathbf{a}, \mathbf{b}), .. d_k(\mathbf{a}, \mathbf{b})) + (1 - \alpha) \cdot \text{mean}(d_1(\mathbf{a}, \mathbf{b}), .. d_k(\mathbf{a}, \mathbf{b})), \tag{28}$$

where $\alpha \in \mathbb{R}$ is an adaptive weight to balance the "max" and "mean" terms. In the scope of our method, we adopt $d_{IQE}(\mathbf{a}, \mathbf{b})$ to implement $\hat{m}$.

### A.2.2 NETWORK ARCHITECTURE

The encoder $\phi$ takes input of the states and consists of 4 convolutional layers followed by 1 fully-connected layer. The output dimension of $\phi$ is 256. The encoder $\psi$ takes input of 512 dimensional vector (concatenated with $\phi(\mathbf{x}_i)$ and $\phi(\mathbf{x}_j)$), feed it into two layer MLPs with 512 hidden units, and output a 256 dim embedding. Q network and policy network are 3-layer MLPs with 1024 hidden units.

### A.2.3 HYPERPARAMETERS

| Hyperparameters | Values |
|---|---|
| Stack frames | 3 |
| Observation shape | $(3 \times 3, 84, 84)$ |
| Action repeat | 2 for finger-spin, walker-walk |
| | 8 for cartpole-swing_up, cartpole-swing_up_sparse |
| | 4 for otherwise |
| Convolutional layers | 4 |
| Convolutional kernal size | $3 \times 3$ |
| Convolutional strides | $[2, 1, 1, 1]$ |
| Convolutional channels | 32 |
| $\phi$ dimension | 256 |
| $\psi$ dimension | 256 |
| Learning rate | 1e-4 |
| Q function EMA $\alpha_Q$ | 0.01 |
| Encoder $\phi$ EMA $\alpha_\phi$ | 0.05 |
| Initial steps | 1000 |
| Replay buffer size | 500K |
| Target update freq | 2 |
| Batch size | 128 |
| Discount factor $\gamma$ | 0.99 |

Table 4: Hyperparameters

### A.2.4 ALGORITHM

---

**Algorithm 1** A learning step in jointly learning SCR and SAC.

---

**Require:** Replay Buffer $\mathcal{D}$, Q network $Q$, policy network $\pi$, target Q network $\bar{Q}$, state encoder $\phi$, target state encoder $\bar{\phi}$, chronological encoder $\psi$.

1: Sample a batch of trajectories with size $B$: $\{\tau_k\}_{k=1}^B \sim \mathcal{D}$
2: Sample state $\mathbf{x}_i$, transition at $\mathbf{x}_i$ and its future state $\mathbf{x}_j$ from each trajectory $\tau_k$: $\{(\mathbf{x}_i, \mathbf{x}_{i+1}, r_i, \mathbf{a}_i, \mathbf{x}_j)_k \sim \tau_k\}_{k=1}^B$
3: Compute loss $\mathcal{L}_\phi(\phi)$ according Equation 2
4: Compute loss $\mathcal{L}_\psi(\psi, \phi)$ according Equation 5
5: Compute loss $\mathcal{L}_{\hat{m}}(\phi)$ according Equation 11
6: Compute loss $\mathcal{L}(\phi, \psi) = \mathcal{L}_\phi(\phi) + \mathcal{L}_\psi(\psi, \phi) + \mathcal{L}_{\hat{m}}(\phi)$ according Equation 12
7: Update $\phi$ and $\psi$ by minimizing loss $\mathcal{L}(\phi, \psi)$
8: Compute RL loss $\mathcal{L}_{RL}$ according to SAC objectives
9: Update $\phi$, $Q$ and $\pi$ by minimizing loss $\mathcal{L}_{RL}$
10: Soft update target Q network: $\bar{Q} = \alpha_Q Q + (1 - \alpha_Q)\bar{Q}$
11: Soft update target state encoder: $\bar{\phi} = \alpha_\phi \phi + (1 - \alpha_\phi)\bar{\phi}$

---

## B  Experiments

### B.1  Additional Information for Distracting Settings in DeepMind Control Suite

In Section 4.2 and 4.4, the evaluating environment has different distracting background videos than training environment.

For the distraction setting in Section 4.2, we utilized the Distracting Control Suite (Stone et al., 2021) with the setting "difficulty=easy". This involves mixing the background with videos from the DAVIS2017 (Pont-Tuset et al., 2017) dataset. Specifically, the training environment samples videos from the DAVIS2017 train set, while the evaluation environment uses videos from the validation set. Each episode reset triggers the sampling of a new video. Additionally, it introduces variability in each episode by applying a uniformly sampled RGB color shift to the robot's body color and randomly selecting the camera pose. The specifics of the RGB color shift range and camera pose variations are in line with the Distracting Control Suite paper (Stone et al., 2021). Different random seeds are used for the training and evaluating environments at the start of training to ensure diverse environments.

For the distraction setting in Section 4.3, we follow the approach in DBC (Zhang et al., 2021) to setup experiments focusing solely on background video distraction. The background videos for this setting are sampled from the Kinetics (Kay et al., 2017) dataset (Kay et al., 2017). We use 1000 consecutive frames for the training environment and a different set of 1000 consecutive frames for the evaluation environment, providing a varied visual context between training and evaluation phases.

### B.2  Additional Experiments on the Number of Steps between $i$ and $j$

In previous experiments, we set the number of steps between $i$ and $j$ to 50. To demonstrate the impact of the hyper-parameter, we include additional experiments with varying step counts: 1, 5, 10, 50, and 100 steps. We evaluate on cheetah-run and walker-walk tasks with the distracting setting in Section 4.2. The results are shown in Figure 6. We observe that that 50 steps yield optimal results in these tasks.

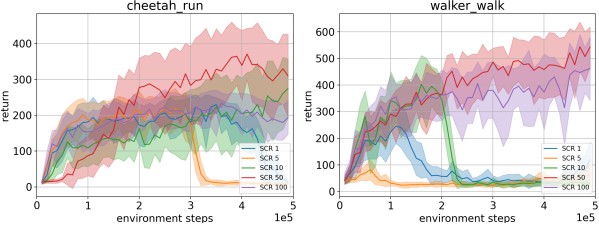

Figure 6: Training curves with varying step counts: 1, 5, 10, 50, and 100 steps in distracting settin. Left: cheetah-run and right: walker-walk. Curves are evaluation scores average on 3 runs and shadow shapes are std.

### B.3  Training Curves of Distracting Control Suite in Section 4.2

Figure 7 shows the training curves of `SCR` and baseline methods in distracting setting in Section 4.2.

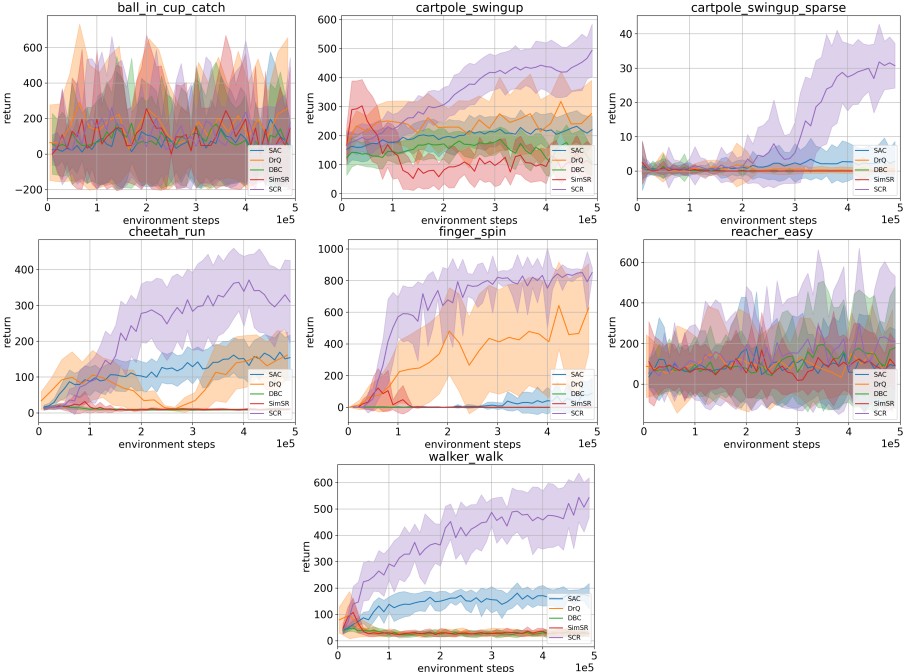

Figure 7: Training curves with varying step counts: 1, 5, 10, 50, and 100 steps in distracting settin. Left: cheetah-run and right: walker-walk. Curves are evaluation scores average on 3 runs and shadow shapes are std.

## B.4 ADDITIONAL EXPERIMENTS ON META-WORLD

In this subsection, we present additional experimental investigations within the Meta-World (Yu et al., 2019), a comprehensive simulated benchmark encompassing 50 distinct robotic manipulation tasks. Our focus narrows to three specific tasks: window-open-v2, door-open-v2, and drawer-open-v2. The observations are rendered as 84 × 84 RGB pixels, consistent to DeepMind Control Suite. The outcomes of these evaluations are depicted in Figure 8. Notably, our proposed SCR outperforms existing baseline methodologies across all evaluated tasks. While the DrQ algorithm demonstrates proficiency in achieving optimal performance levels, it is observed that SCR maintains superior sample efficiency, underscoring its effectiveness in the applied setting.

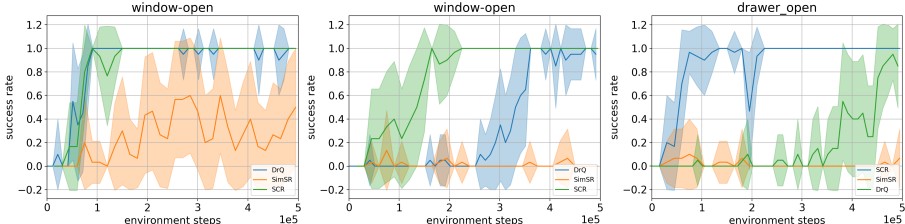

Figure 8: Training Curves of Meta-World. From left to right: window-open-v2, door-open,-v2 drawer-open-v2. Curves are evaluation scores average on 3 runs and shadow shapes are std.

