# OpenReview forum: "State Chrono Representation for Enhancing Generalization in Reinforcement Learning"
_ICLR.cc/2024/Conference — ICLR 2024 Conference Withdrawn Submission_

### Official Review · Reviewer_JyET · 2023-10-30

**Soundness:** 2 fair
**Presentation:** 2 fair
**Contribution:** 2 fair
**Rating:** 3
**Confidence:** 5

**Summary:**

This paper proposes a method to learn state representation based on bisimulation-like measurement for deep reinforcement learning. The whole model consists of three modules: i) a bisimulation-based module to capture behavior differences between states, which includes a novel diffuse metric; ii) a chronological embedding module to capture long-term temporal information; and iii) a temporal measurement module to capture discounted cumulative reward of sub-trajectories. The empirical results show their performance improvements over the previous representation approaches, not only in default settings but also in more challenging distractor settings.

**Strengths:**

This paper is easy to follow. The authors provide experimental evidence to support the performance improvement of the proposed method on different variations of DeepMind Control Suite.

**Weaknesses:**

Many of the statements and explanations in this article seem to be questionable, which consequently lead to a decrease in the overall soundness of the paper. I will now enumerate some of these problems:

1. In the fourth paragraph of the introduction section, "*behavioral metrics only capture immediate, one-step information, neglecting the broader long-term context*". However, behavioral metrics consider both immediate rewards and long-term behavioral differences, instead of only capturing one-step information. This is the reason why bisimulation was initially chosen to learn state representations. The poor performance of behavioral metrics in sparse environments is not due to a lack of consideration of long-term information but rather because the supervision signal provided by rewards is too limited in sparse reward settings, which makes updates challenging. This issue is similar to the value iteration in sparse reward environments.

2. In the paragraph after Definition 3.1, "*SimSR utilizes the cosine distance, derived from the cosine similarity, albeit
without fulfilling the triangle inequality and the non-zero self-distance*". In fact, cosine distance is indeed zero self-distance as the fact that $d(u, u) = 0$.

3. In the first sentence of Section 3.1.1, "*The loss $L_\phi(\phi)$ in Equation 2 leans heavily on a one-step update mechanism*". It should be referred to as **temporal-difference update mechanism** rather than a **one-step update mechanism**.

4. The proof for Theorem 5 appears somewhat ambiguous. Indeed, if we assume that $x_j$ and $y_j'$ are fixed states, then this proof seems to be reasonable. However, upon the algorithm in Appendix A.2.4, it appears that neither of these states is actually fixed. Consequently, the proof for Theorem 5 is entirely different from that of Theorem 3. One potential solution could be to see the combination of $x_i$ and $x_j$ as one auxiliary state and the combination of $y_i' $and $y_j'$ as another auxiliary state, then iterate the bisimulation operator in the auxiliary state space. However, in this setting, the mapping from the state space to the auxiliary state space would be dimensionality-increasing, which is contrary to our intended goal. Therefore, I'm somewhat uncertain about the validity of this theorem.

5. Equation 9 also seems to be strange. $m$ should represent the cumulative rewards under the optimal policy. However, on the right-hand-side of Equation 9, the policy associated with $d$ is somewhat unclear or uncertain. i) If it is the current policy during the optimization, then it could possibly be zero initially. In extreme cases where the initial value of $m$ is the same and sufficiently large for all state pairs, it's possible that, under the conditions of Equation 9, Equation 8 could be consistently equal to 0, leading to a situation where $m$ does not get updated. ii) If the policy on the right-hand-side of Equation 9 is the optimal policy, then as the optimal policy is unknown, we also cannot obtain the corresponding $d^{π^*}$. In such cases, Equation 10 would indeed not hold.

6. I would like to recommend that the authors include a performance curve to illustrate the trend. Furthermore, providing the code would enhance the soundness of the paper.

Minor issue:

1. Legend in Figure 5:  "CR" -> "SCR"

2. Figure 6 -> Table 3

**Questions:**

1. What is the future state? How can we sample it? Is it randomly selected from uniform distribution or what?

2. In the Experiment section, does the "*500K step*" refer to environment steps or gradient steps?

3. How does it perform if we only use $\mathcal{L}_\phi(\phi)$?

4. How many random seeds are used in Table 2 and Figure 6?

5. What was the experimental setup for distraction settings? Are the training environment and the evaluating environment the same or different?

---

> ### Author Response · Authors · 2023-11-22
> **Response to Reviewer JyET (1/3)**
>
> We appreciate the detailed feedback provided by the reviewer and have revised our manuscript accordingly. We address reviewer's concern as follows.
>
> > In the fourth paragraph of the introduction section, "behavioral metrics only capture immediate, one-step information, neglecting the broader long-term context". However, behavioral metrics consider both immediate rewards and long-term behavioral differences, instead of only capturing one-step information. This is the reason why bisimulation was initially chosen to learn state representations. The poor performance of behavioral metrics in sparse environments is not due to a lack of consideration of long-term information but rather because the supervision signal provided by rewards is too limited in sparse reward settings, which makes updates challenging. This issue is similar to the value iteration in sparse reward environments.
>
> We appreciate your insightful comment regarding behavioral metrics and their consideration of long-term behavioral information. To clarify, while behavioral metrics do indeed take into account long-term behavioral differences, the mechanism of their update primarily focuses on one-step transitions. This, particularly in sparse reward environments, limits the amount of information each update can leverage. Our method, as delineated in Equations (5) and (11), addresses this limitation by incorporating updates that encompass long-term behaviors, including n-step transitions. This approach aligns with strategies used in conventional deep reinforcement learning methods such as DQN and Rainbow, where n-step updates have been proven effective in enhancing performance in sparse reward settings.
>
> By integrating n-step transition information, our method is designed to more rapidly assimilate behavioral differences, thus effectively navigating the challenges posed by sparse reward environments.
>
>
> > In the paragraph after Definition 3.1, "SimSR utilizes the cosine distance, derived from the cosine similarity, albeit without fulfilling the triangle inequality and the non-zero self-distance". In fact, cosine distance is indeed zero self-distance as the fact that $d(u, u)=0$.
>
> Thank you for pointing out the potential confusion in our description of cosine distance in SimSR. Our statement claimed that SimSR is not non-zero self-distance, which indicates SimSR has zero self-distance. We will revise the text to clearly convey this. Additionally, we note that cosine distance has zero self-distance, this characteristic is **not good** for sample-based behavioral metrics, because MICo(Castro et al. 2021) mentioned that sample-based behavioral metrics are Łukaszyk-Karmowski distance which has non-zero self-distance.
>
> > In the first sentence of Section 3.1.1, "The loss $L_\phi(\phi)$ in Equation 2 leans heavily on a one-step update mechanism". It should be referred to as temporal-difference update mechanism rather than a one-step update mechanism.
>
> Thank you for the suggestion. We have revised the sentence in Section 3.1.1 to accurately reflect that the loss $L_\phi(\phi)$  in Equation 2 leans heavily on temporal-difference update mechanism with one-step transitions information.
>
>
> > The proof for Theorem 5 appears somewhat ambiguous. Indeed, if we assume that $x_j$ and $y'_j$ are fixed states, then this proof seems to be reasonable. However, upon the algorithm in Appendix A.2.4, it appears that neither of these states is actually fixed. Consequently, the proof for Theorem 5 is entirely different from that of Theorem 3. One potential solution could be to see the combination of $x_i$ and $x_j$ as one auxiliary state and the combination of $y'_i$ and $y'_j$ as another auxiliary state, then iterate the bisimulation operator in the auxiliary state space. However, in this setting, the mapping from the state space to the auxiliary state space would be dimensionality-increasing, which is contrary to our intended goal. Therefore, I'm somewhat uncertain about the validity of this theorem.
>
> Thank you for your insights regarding the proof of Theorem 5. In addressing your concerns, we propose that combining $x_i$ and $x_j$ into a single auxiliary state allows for a reformulation of Theorem 5 in a manner akin to Theorem 3. This do not violate the proof of Theorem 5. While this approach does indeed increase the dimensionality of the auxiliary state space, it does not contradict our paper's goal. Our primary goal is to map high-dimensional pixel states to lower-dimensional representations. The encoder $\psi$ effectively transforms these enlarged auxiliary states into compact embeddings. The dimensions of $\phi$ and $\psi$, both set to 256, are detailed in Appendix A.2.3, Table 4. This approach ensures the validity of Theorem 5 within the scope of our methodology.

---

> ### Author Response · Authors · 2023-11-22
> **Response to Reviewer JyET (2/3)**
>
> > Equation 9 also seems to be strange. $m$ should represent the cumulative rewards under the optimal policy. However, on the right-hand-side of Equation 9, the policy associated with $d$ is somewhat unclear or uncertain. i) If it is the current policy during the optimization, then it could possibly be zero initially. In extreme cases where the initial value of $m$ is the same and sufficiently large for all state pairs, it's possible that, under the conditions of Equation 9, Equation 8 could be consistently equal to 0, leading to a situation where $m$ does not get updated. ii) If the policy on the right-hand-side of Equation 9 is the optimal policy, then as the optimal policy is unknown, we also cannot obtain the corresponding $d^{\pi^{*}}$. In such cases, Equation 10 would indeed not hold.
>
> Thank you for your observations regarding Equation 9. We acknowledge that Equation 9, representing an inequality, is proposed based on intuition, particularly reflecting the triangle inequality concept as depicted in Figure 2. It's not formulated as a theorem due to its nature, which is not amenable to theoretical proof. Instead, it serves as an upper boundary in our deep learning framework to prevent the divergence of training for $\hat{m}$.
>
> **Concern i)**: $d$ is indeed associated with the current policy. However, due to the random initialization of neural networks, the initial value of $d$ is not zero but a small value. Additionally, both $d$ and $\hat{m}$ are non-parametric functions computed based on $\phi{x}$, as detailed in Definition 3.2 and Appendix A.2.1, respectively. The representation $\phi{x}$ is learned through multi-task objectives, including $d$, $\hat{m}$ , and the SAC objectives (updating the Q-function and policy). This intricate training setup ensures that both $d$ and $\hat{m}$ are not likely to converge to zero.
>
> **Concern ii)**: The $\hat{m}$ in Equation 9 refers to the on-policy $d^{\pi}$. Our implementation assumes that the approximation $\hat{m}$ will closely align with the true $m$ as the policy $\pi$ approaches optimality. When $\pi$ is suboptimal, a range for $\hat{m}$, bounded by Equations 7 and 9, still exists. This range dynamically adjusts during training, allowing $\hat{m}$ to approximate the true $m$ as $\pi$ becomes more optimal.
>
> > I would like to recommend that the authors include a performance curve to illustrate the trend. Furthermore, providing the code would enhance the soundness of the paper.
>
> We have added detailed training curves for experiments in Section 4.3, which can now be found in Appendix B.3 of the latest version of our paper.
> We will open source code once this paper is accepted.
>
> > What is the future state? How can we sample it? Is it randomly selected from uniform distribution or what?
>
> The concept of a 'future state' in our framework is derived from the sampling of state in the replay buffer. Specifically, we uniformly sample a trajectory from the replay buffer and then select two states within this trajectory. The state that chronologically follows the other, denoted as $x_j$, is treated as the 'future state.' We have detailed this sampling procedure in Algorithm 1, presented in Appendix A.2.4, lines 1-2 of our paper.
>
> > In the Experiment section, does the "500K step" refer to environment steps or gradient steps?
>
> The term "500K step" in our experiment section refers to 500K environment interactions. To elaborate, for every environment step, we perform one gradient step. However, for the policy network, we take a gradient step after every two environment steps.
>
> > How does it perform if we only use $L_\phi(\phi)$ ?
>
> We have conducted an updated ablation study to evaluate the performance when using only $L_\phi(\phi)$. This study is now included in Figure 5 in Section 4.3 of the latest version of our paper. The results indicate that using only $L_\phi(\phi)$ yields suboptimal outcomes compared to our full method. Specifically, for the walker-walk task, the performance was significantly lower, similar to that of DBC and SimSR. In contrast, for the cheetah-run task, the results were comparable to those achieved with SAC and DrQ. These findings underscore the importance of our full model.
>
> > How many random seeds are used in Table 2 and Figure 6?
>
> For all our experiments, including those presented in Table 2 and Figure 6, we employed 3 random seeds. This detail will be explicitly stated in the latest version of our paper to provide clarity on the robustness and reliability of our experimental results.

---

> ### Author Response · Authors · 2023-11-22
> **Response to Reviewer JyET (3/3)**
>
> > What was the experimental setup for distraction settings? Are the training environment and the evaluating environment the same or different?
>
> In our experiments, the distracting background videos in the training environment differ from those in the evaluation environment to ensure a robust assessment of our model's generalizability.
>
> **Distraction Setting in Section 4.2:**
> We utilized the Distracting Control Suite (Stone et al., 2021) with the setting "difficulty=easy". This involves mixing the background with videos from the DAVIS2017 dataset. Specifically, the training environment samples videos from the DAVIS2017 train set, while the evaluation environment uses videos from the validation set. Each episode reset triggers the sampling of a new video. Additionally, it introduces variability in each episode by applying a uniformly sampled RGB color shift to the robot's body color and randomly selecting the camera pose. The specifics of the RGB color shift range and camera pose variations are in line with the Distracting Control Suite paper (Stone et al., 2021). Different random seeds are used for the training and evaluating environments at the start of training to ensure diverse environments.
>
> **Distraction Setting in Section 4.3:**
> Following the approach in DBC (Zhang et al., 2021), our experiment in this section focuses solely on background video distraction. The background videos for this setting are sampled from the Kinetics dataset (Kay et al., 2017). We use 1000 consecutive frames for the training environment and a different set of 1000 consecutive frames for the evaluation environment, providing a varied visual context between training and evaluation phases.
>
> To provide comprehensive details on these settings, we have added a dedicated section in Appendix B.1 of our paper.
>
> --------------
> Reference
>
> (Kay et al. 2017) The kinetics human action video dataset. CoRR. 2017.

---

### Official Review · Reviewer_yyec · 2023-10-30

**Soundness:** 3 good
**Presentation:** 3 good
**Contribution:** 3 good
**Rating:** 8
**Confidence:** 3

**Summary:**

The paper introduces a new representation learning approach that can be seamlessly integrated with any reinforcement learning algorithm. Central to this method is the enhancement of traditional bisimulation/MICo metrics [0].
The authors innovatively incorporate long-term information into the representation. This is achieved by applying a form of MICo distance on the joint embeddings of the representations of both the initial and terminal states of sub-trajectory pairs.
Additionally, they introduce a *temporal measurement* which quantifies the difference in values between these states, further enriching the long-term information within the representations.
Unlike conventional bisimulation or MICo metrics that predominantly capture immediate one-step relations and often overlook long-term dynamics, these added objectives ensure a holistic capture of both immediate and long-term behaviors.
Empirical validations demonstrate the method's efficacy, as it showcases robust performance in learning optimal policies on the dm-control suite benchmark. Moreover, the approach demonstrates impressive generalization abilities in test scenarios with distracting elements.

[0] MICo: Improved representations via sampling-based state similarity for Markov decision processes
Pablo Samuel et al. Neurips 2021

**Strengths:**

The paper is clearly written, making it relatively easy to follow for readers.
The authors provide a sound motivation behind each component of the method they introduce.
They offer a modified distance for the MICo metric, which simplifies its implementation and could improve its numerical stability.
An ablation study is included, highlighting the effects of individual components of the method.
Notably, the method demonstrates good performance and an ability to handle distracting factors, even though it doesn't have explicit mechanisms to counteract these distractions.

**Weaknesses:**

Certain concepts introduced in the paper, specifically the Łukaszyk-Karmowski distance and the Interval Quasimetric Embedding, would benefit from further elucidation for clarity. On the implementation front, several pertinent details are lacking, which could hinder attempts at reproducing the method faithfully. For instance, in computing the *chrono embeddings* $\psi(x_i, x_j)$, the method doesn't specify how the number of steps between $i$ and $j$ are determined. Given the potential significance of this hyper-parameter on the chronological behavioral metric, addressing its determination and providing further discussion would have added depth to the paper.

Additionally, the role and necessity of the encoder $\phi$ EMA parameter isn't evident. While one might speculate it's used for targets in $L_\phi$ and $L_\psi$​, explicit clarification is absent.

A major concern revolves around the method's adaptability to various RL environments. The paper describes chrono embeddings based on MICo metrics over trajectory pairs, suitable for capturing long-range similarities in deterministic settings, such as the DM-control environments. However, its behavior in stochastic environments remains questionable. For example, in games like Mario, two visually identical states from separate levels might seem alike using bisimulation metrics, but the proposed chronological behavioral metric could consider them as vastly dissimilar due to the ensuing trajectory differences.
The inclusion of experimental validation on varied environments, such as procgen, would further enhance the paper by addressing these concerns comprehensively.

**Minor issues observed:**

*    In Theorem 2, the end of the equation should reference $d(x',y')$ rather than $d(x,y)$.
*    There seems to be inconsistency with terminologies, as the DBC is mentioned between definitions 3.1 and 3.2 without prior introduction.
*    Theorem 5 should align with the subsequent explanation by utilizing $d^{\pi}_\phi$.
*    A typo is present post equation 4 with "TThe encoder..."
*    In Figure 5, the left figure's legend incorrectly labels two types as CR, when one should be SCR.

**Questions:**

* Can the authors discuss about the impact of the number of steps between $i$ and $j$ when computing the *chrono embeddings*
* Are the gradient stopped for computing the targets used in $L_\phi$ and $L_\psi$​ or are they computed with a target encoder?
* Could the authors measure the impact of the introduced diffused metric on MICo for exemple in the ablation study.
It would help to assess its individual contribution.

**Details Of Ethics Concerns:**

* Could the authors elaborate on the impact of the number of steps between \( i \) and \( j \) in the computation of the *chrono embeddings*? How is this value determined and what influence does it have on the overall method?
* For the targets used in \( L_\phi \) and \( L_\psi \), are the gradients halted during computation or is a target encoder employed for this purpose?
* Would it be possible for the authors to evaluate the effect of the introduced diffused metric on MICo, perhaps within the framework of the ablation study? Understanding its distinct contribution could provide deeper insights into the overall performance of the method.

---

> ### Author Response · Authors · 2023-11-22
> **Response to Reviewer yyec (1/2)**
>
> Thank you for your constructive comments. We address each of the concerns raised.
>
> > Certain concepts introduced in the paper, specifically the Łukaszyk-Karmowski distance and the Interval Quasimetric Embedding, would benefit from further elucidation for clarity.
>
> The Łukaszyk-Karmowski distance is a measure between two random variables, represented as $D_{LK}(d)(X,Y)=\int \int d(x, y) f(x) g(y) dx dy$, where $f(x)$ and $g(y)$ are the probability density functions of $X$ and $Y$ respectively, and $d$ is a metric. This can also be expressed as $D_{LK}(d)(X,Y)= \mathbb{E}_{x \sim X, y \sim Y} [d(x, y)]$. Note that Łukaszyk-Karmowski distance doesn't satisfy the identity of indiscernibles, which is why it is not considered a metric.
>
> The Interval Quasimetric Embedding (IQE), introduced by (Wang & Isola, 2022a), offers a novel approach to computing quasi-metrics between vector embeddings, which are inherently asymmetric, i.e., $d(a,b) \neq d(b,a)$. IQE divides each vector into $k$ segments and computes distances within these segments by laying out each dimension on a real line. The overall distance is then calculated using a combination of "max" and "mean" operations across these component distances. Further details are elaborated in Appendix A.2.1.
>
> > Additionally, the role and necessity of the encoder $\phi$ EMA parameter isn't evident. While one might speculate it's used for targets in $L_\phi$ and $L_\psi$​, explicit clarification is absent.
>
> We acknowledge the need for clarification regarding the encoder $\phi$ EMA parameter. This parameter is not utilized in our proposed losses $L_\phi$, $L_\psi$ and $L_{\hat{m}}$. Instead, the target encoder $\bar{\phi}$ is computed for the target Q-function $\bar{Q}$, as the Q-network relies on inputs from $\phi(x)$. This approach is in line with previous RL works, such as SAC with pixel encoder, DrQ, and DBC. Our paper employs a similar target encoder network, which is softly updated alongside the target Q-network.
>
> > A major concern revolves around the method's adaptability to various RL environments. The paper describes chrono embeddings based on MICo metrics over trajectory pairs, suitable for capturing long-range similarities in deterministic settings, such as the DM-control environments. However, its behavior in stochastic environments remains questionable. For example, in games like Mario, two visually identical states from separate levels might seem alike using bisimulation metrics, but the proposed chronological behavioral metric could consider them as vastly dissimilar due to the ensuing trajectory differences.
> The inclusion of experimental validation on varied environments, such as procgen, would further enhance the paper by addressing these concerns comprehensively.
>
> To address this issue, we have included additional experiments in the Meta-World environment, encompassing 50 robotic manipulation tasks. Specifically, we focused on three tasks: window-open, door-open, and drawer-open. Our results, shown in Appendix B.3, demonstrate superior performance compared to baseline methods in these tasks. While we acknowledge the importance of testing in environments like Procgen, the time constraints of the discussion period limited our ability to conduct these experiments. However, we plan to explore this in our future work.
>
> Regarding the issue in the Mario game, it seems to stem from the POMDP nature of the environment, where frames do not fully represent the true states. Our method, assuming fully observable environments, might not directly address this. We recognize the significance of POMDPs and intend to explore this in future research.
>
> > On the implementation front, several pertinent details are lacking, which could hinder attempts at reproducing the method faithfully. For instance, in computing the chrono embeddings $\psi(x_i, x_j)$, the method doesn't specify how the number of steps between $i$ and $j$ are determined. Given the potential significance of this hyper-parameter on the chronological behavioral metric, addressing its determination and providing further discussion would have added depth to the paper.
>
> > Can the authors discuss about the impact of the number of steps between $i$ and $j$ when computing the chrono embeddings.
>
> We thank you for highlighting the need for additional details on implementation. The number of steps between $i$ and $j$ in our experiments is consistently set to 50. To demonstrate the impact of this parameter, we have included additional experiments with varying step counts: 1, 5, 10, 50, and 100 steps. These experiments, detailed in Appendix B.3, reveal that 50 steps yield optimal results in tasks like cheetah-run and walker-walk under the Distracting Control Suite setting.

---

> ### Author Response · Authors · 2023-11-22
> **Response to Reviewer yyec (2/2)**
>
> > Minor issues observed:
> >* In Theorem 2, the end of the equation should reference $d(x',y')$ rather than $d(x, y)$
> >* There seems to be inconsistency with terminologies, as the DBC is mentioned between definitions 3.1 and 3.2 without prior introduction.
> >* Theorem 5 should align with the subsequent explanation by utilizing $d^\pi_\psi$
> >* A typo is present post equation 4 with "TThe encoder..."
> >* In Figure 5, the left figure's legend incorrectly labels two types as CR, when one should be SCR.
>
> We are grateful for the identification of these minor issues. We revise accordingly in the latest version of paper.
>
> > Are the gradient stopped for computing the targets used in $L_\phi$ and $L_\psi$​ or are they computed with a target encoder?
>
> The gradients for computing targets in $L_\phi$ and $L_\psi$ are not impacted by the target encoder or target Q-network. These are exclusively computed in the TD update for the Q-function, following the SAC algorithm. The soft updates for the target encoder and Q-network are detailed in Algorithm 1, Appendix A.2.4.
>
> > Could the authors measure the impact of the introduced diffused metric on MICo for example in the ablation study. It would help to assess its individual contribution.
>
> Our ablation study in Section 4.3 addresses this query. Specifically, Figure 5 compares our method with variations using different distance metrics: cosine distance (SimSR), angular distance (MICo), and L1 distance. The results indicate that while our method with MICo achieves comparable performance in the cheetah-run task, it struggles in the walker-walk task. This comparison underscores the unique contribution and impact of our diffused metric on the overall methodology.
>
>
> --------
> Reference
>
> (Wang & Isola, 2022a). Tongzhou Wang et al. Improved representation of asymmetrical distances with interval quasimetric embeddings. NeurIPS Workshop. 2022.

---

### Official Review · Reviewer_AZZ2 · 2023-11-01

**Soundness:** 2 fair
**Presentation:** 2 fair
**Contribution:** 2 fair
**Rating:** 5
**Confidence:** 4

**Summary:**

The authors propose a method, the State Chrono Representation (SCR), integrating long-term information alongside the bisimulation metric learning.
They tackle with the problem of a robust and generalizable state representation, especially the issue of long-term behaviors within their representations.
The deep bisimulation metric approaches transforms states into structured representation spaces, allowing the measurement of distances based on task-relevant. However, in sparse rewards scenarios, due to its one-step update approach the generalized state issue still remains.
Capturing long-term behaviors within their representations are expected.
To solve this problem, they SCR approach, integrating long-term information alongside the bisimulation metric.
They showed that SCR has comparable results with the augmentation method DrQ and the state-of-the-art behavioral metric approach SimSR.

**Strengths:**

As the authors point out, capturing long-term behaviors is a key issue in RL.
To fuse temporal relevance to metric learning is a straight forward and interesting approach.
SCR shows better performance in some cases than SOTA methods about sample-efficiency and generalization with corresponding experiments.
They also clarify the contribution of each component by removing some components of SCR that clarify the detailed analysis and contributions of SCR.

**Weaknesses:**

Performance comparisons vary depend on tasks.
Some are better and some are worse.
Analysis of are limited and the insights into reasons for those better and worse performances are not enough to support their claims.

**Questions:**

The main issue of the paper is to solve the problem of long-term behaviors within their representations.
Experimental analysis are only from the viewpoint of sample efficiency and generalization, but few analysis about the long-term behaviors. With additional analysis from the long-term behavior  views would support the authors' claims more.

---

> ### Author Response · Authors · 2023-11-22
> **Response to Reviewer AZZ2**
>
> Thank you for your constructive comments. We appreciate the opportunity to clarify and elaborate on our paper's motivation, contributions, and experimental analysis.
>
> Our paper aims to develop a robust and generalizable metric-based state representation, enhancing generalization and efficiency in reinforcement learning (RL) tasks. We identified that existing metric-based state representation methods, despite their remarkable performance, struggle with capturing long-term behaviors. This shortfall results in diminished efficiency and generalization in challenging RL tasks, particularly those with non-informative rewards such as sparse rewards. To address this, we introduced the SCR approach, designed to effectively capture long-term behaviors, thereby improving generalization, efficiency, and performance in sparse reward RL tasks.
>
> > Performance comparisons vary depend on tasks. Some are better and some are worse. Analysis of are limited and the insights into reasons for those better and worse performances are not enough to support their claims.
>
> We appreciate the opportunity to provide further insights into our experimental results and their significance:
>
> **Distracting Control Suite (Section 4.2):** Our main experimental focus, the Distracting Control Suite, is intentionally challenging, incorporating natural video backgrounds, randomized robot colors, and varying camera poses. In this complex setting, our method, SCR, outperformed baseline methods in 6 out of 7 tasks. The exception, "ball_in_cup-catch," is inherently a sparse reward task and its difficulty is amplified with distractions. The high variance observed in the scores of this task results in overlapping confidence intervals between SCR and DrQ, but this should not detract from the overall effectiveness of our method in such challenging conditions.
>
> **Default Settings of DM_Control (Section 4.1):**  In this experiment, conducted under the default settings of DM_Control with static backgrounds and fixed camera viewpoints, it is important to note that the optimal performance score is 1000. Existing methods in these settings typically achieve scores in the 800-900 range, which are close to this optimal benchmark. This experiment was crucial to showcase that SCR does not sacrifice generalization for peak performance. In this standard setting, the scores of SCR closely mirror those of the top-performing baseline methods, indicating its robustness and efficiency.
>
> **Distracting Setting with Fixed Camera Pose (Section 4.3):** In a variant of the distracting setting, where only background videos are altered, SCR was re-evaluated with an adjusted learning rate (1e-3) in the revised version of paper. In this setting, SCR surpassed the performance of the state-of-the-art method, SimSR, in 6 out of 7 tasks. The enhanced performance of SimSR compared to Section 4.2 indicates that a randomized camera pose is a significant factor in observation distraction. The updated scores in our latest manuscript version reflect these findings.
>
> **Meta-World benchmark**: We extended our evaluation to the Meta-World benchmark (Appendix B.4) in the revised version of the paper, which includes 50 robotic manipulation tasks. Our method demonstrated superior performance in tasks like window-open, door-open, and drawer-open, further validating its effectiveness across diverse RL domains.
>
> > The main issue of the paper is to solve the problem of long-term behaviors within their representations. Experimental analysis are only from the viewpoint of sample efficiency and generalization, but few analysis about the long-term behaviors. With additional analysis from the long-term behavior views would support the authors' claims more.
>
> The primary focus of our work is developing a robust metric-based state representation, emphasizing sample efficiency and generalization. However, addressing long-term behaviors is indeed central to our motivation. Our experimental results, particularly in tasks with sparse rewards and challenging generalization scenarios, demonstrate that SCR can effectively capture long-term behaviors, as evidenced by its superior performance in these settings.

---

### Official Review · Reviewer_w1s3 · 2023-11-01

**Soundness:** 3 good
**Presentation:** 2 fair
**Contribution:** 2 fair
**Rating:** 3
**Confidence:** 3

**Summary:**

This paper proposes a set of losses for effectively learning representations for efficient downstream RL.

**Strengths:**

- This paper studies an interesting problem of learning effective representations for reinforcement learning
- This paper proposes a variety of different losses for temporal representation learning

**Weaknesses:**

- The writing seems to be overly grandiose, for instance in the conclusion "Our novel metric-based framework, State Chrono
Representation (SCR), emerges as a beacon in this realm, drawing on the temporal essence of RL.".

- Several of the theorems in the paper aren't theorem but definitions -- the authors should rename the statements as such

- The writing of the theorems are not fully clear -- for instance, in Thererom 1, A is never defined

- The different losses proposed in the paper appear to be a bit ad-hoc to me -- more justification on the precise forms used would be helpful

- The results in the paper are not convincing, many of the confidence intervals overlap with each other

- The evaluation in the paper seems limited, it is only evaluated on different variations of Mujoco -- it would be good to provide other results in domains such as RL-Bench, Minecraft, Antmaze etc...

**Questions:**

1) Can the author elaborate in the related work how this differs from prior works in the representation learning space?

---

> ### Author Response · Authors · 2023-11-22
> **Response to Reviewer w1s3 (1/2)**
>
> We appreciate the reviewer's time and insightful feedback. Below, we address each of the concerns raised:
>
> > The writing seems to be overly grandiose, for instance in the conclusion "Our novel metric-based framework, State Chrono Representation (SCR), emerges as a beacon in this realm, drawing on the temporal essence of RL.".
>
> Thank you for pointing out the overly grandiose style in our conclusion. We have revised it to be more precise and less embellished: "In this work, we propose a novel metric-based representation framework State Chrono Representation (SCR) that leverages temporal dynamics in RL."
>
> > Several of the theorems in the paper aren't theorem but definitions -- the authors should rename the statements as such
>
> We appreciate the reviewer’s attention to the distinction between theorems and definitions in our paper. We offer the following clarifications:
>
> We introduced Theorems 3, 4, and 5, while Theorems 1 and 2 are based on prior works by (Zhang et al., 2021) and (Castro et al., 2021),respectively, and are referenced in the preliminary section for context.
>
> **Fixed-Point in Behavioral Metrics**: Theorems 3 and 5  assert that the behavioral metrics update operator we have defined has fixed-point. We substantiate the existence of these fixed-point in Appendices A.1.1 and A.1.3. The significance of fixed-point, as per Banach's fixed-point theorem, lies in the convergence of a distance function $d$ when iteratively updated by a behavioral metric update operator $\mathcal{F}$. This convergence is crucial; without a fixed-point (or convergence guarantee), it becomes impractical to approximate such a distance function with a neural network. This underscores why fixed-point existence is vital for defining a behavioral metric.
>
> Besides, many papers defined the update operator $\mathcal{F}$ in our theorems aligns with approaches in other papers, such as (Castro, 2020) and (Ferns & Precup, 2014). We don't separate definition and theorem in order to save page space.
>
> **Theorem 4** establishes that $\hat{d}$ is a diffuse metric, where $\hat{d}$ is detailed in Definition 3.2 and the concept of a diffuse metric is outlined in Definition 3.1. The validity of Theorem 4 is proved in Appendix A.1.2.
>
> > The writing of the theorems are not fully clear -- for instance, in Thererom 1, A is never defined
>
> Regarding the specific concern in Theorem 1, $\mathcal{A}$ represents the action space within the Markov Decision Process (MDP), and this is indeed defined in Section 2, under the paragraph 'Markov Decision Process', on Page 2. We recognize the importance of clear and accessible notation and will ensure all such notations are explicitly and clearly defined in the revised version of paper.
>
> > The different losses proposed in the paper appear to be a bit ad-hoc to me -- more justification on the precise forms used would be helpful
>
> We appreciate the opportunity to elaborate on the loss functions introduced in our paper.
>
> **$L_\phi$ (in Equation 2)** is designed to learn the behavioral metric as defined in Theorem 3. The learning of behavioral metrics is a proven approach for efficient state representation in RL. What sets $L_\phi$ apart is its basis on a new distance function $\hat{d}$, specified in Definition 3.2. This function, a diffuse metric, is not only novel in our work but also straightforward to implement. This approach draws inspiration from, yet advances beyond, similar loss functions found in DBC and SimSR.
>
> **$L_\psi$ (in Equation 5)** is to learn the chronological embedding $\psi(x_i, x_j)$, which approximates the metric $d_\psi$ as defined in Theorem 5. Here, $\psi$captures the temporal relationship between a current state
> $x_i$  and its subsequent long-term states $x_j$. This is a novel component of our framework, as it delves into the temporal dynamics of states in RL.
>
> In Section 3.3, we introduce a **temporal measurement $\hat{m}$** to quantify the discrepancies between current and future states. Unlike traditional regression loss approaches, we have innovatively designed lower and upper boundaries for
> $\hat{m}$, ensuring that it remains within a feasible and meaningful range. These boundaries are reflected in the losses presented in Equations 7 and 9.
>
> All these losses are not trained in isolation but are integrated and trained jointly as delineated in Equation 12.

---

> ### Author Response · Authors · 2023-11-22
> **Response to Reviewer w1s3 (2/2)**
>
> > The results in the paper are not convincing, many of the confidence intervals overlap with each other
>
> We appreciate the opportunity to provide further insights into our experimental results and their significance:
>
> **Distracting Control Suite (Section 4.2):** Our main experimental focus, the Distracting Control Suite, is intentionally challenging, incorporating natural video backgrounds, randomized robot colors, and varying camera poses. In this complex setting, our method, SCR, outperformed baseline methods in 6 out of 7 tasks. The exception, "ball_in_cup-catch," is inherently a sparse reward task and its difficulty is amplified with distractions. The high variance observed in the scores of this task results in overlapping confidence intervals between SCR and DrQ, but this should not detract from the overall effectiveness of our method in such challenging conditions.
>
> **Default Settings of DM_Control (Section 4.1):**  In this experiment, conducted under the default settings of DM_Control with static backgrounds and fixed camera viewpoints, it is important to note that the optimal performance score is 1000. Existing methods in these settings typically achieve scores in the 800-900 range, which are close to this optimal benchmark. This experiment was crucial to showcase that SCR does not sacrifice generalization for peak performance. In this standard setting, the scores of SCR closely mirror those of the top-performing baseline methods, indicating its robustness and efficiency.
>
> **Distracting Setting with Fixed Camera Pose (Section 4.3):** In a variant of the distracting setting, where only background videos are altered, SCR was re-evaluated with an adjusted learning rate (1e-3) in the revised version of paper. In this setting, SCR surpassed the performance of the state-of-the-art method, SimSR, in 6 out of 7 tasks. The enhanced performance of SimSR compared to Section 4.2 indicates that a randomized camera pose is a significant factor in observation distraction. The updated scores in our latest manuscript version reflect these findings.
>
> > The evaluation in the paper seems limited, it is only evaluated on different variations of Mujoco -- it would be good to provide other results in domains such as RL-Bench, Minecraft, Antmaze etc...
>
> To address the limitations noted, we have conducted new experiments in **Meta-World**, a simulated robotic manipulation environment akin to RL-Bench. This environment presents a diverse range of 50 robotic manipulation tasks. We specifically selected three tasks for evaluation: window-open, door-open, and drawer-open. The results of these experiments, detailed in Appendix B.4, demonstrate that our method outperforms other baseline methods in these varied tasks. This addition significantly broadens the scope of our evaluation and illustrates the adaptability of our method to different contexts.
>
> Regarding the Antmaze benchmark, it is pertinent to note that it is typically associated with offline RL. In the context of online RL, Antmaze presents a significant challenge in terms of hard exploration. Most current online RL approaches, including ours, struggle with such hard exploration problems without specific exploration strategies.
>
> While we acknowledge the potential of Minecraft as a benchmark for evaluating RL methods, due to the time constraints of the rebuttal period, we were unable to conduct experiments in this domain. However, we recognize its importance and relevance and plan to include Minecraft in our future work to further validate and expand the applicability of our approach.
>
> > Can the author elaborate in the related work how this differs from prior works in the representation learning space?
>
> Our work, while employing a behavioral metric akin to MICo (Castro et al., 2021) and SimSR (Zang et al., 2022) for learning the encoder $\phi$, distinguishes itself in two significant ways: Firstly, SCR introduces a novel form of approximation for the encoder, as detailed in Definition 3.2. Secondly, we expand the representation learning scope by incorporating a state-pair encoder $\psi$ to capture long-term behavioral state relationships and an additional component $\hat{m}$ to enhance encoder $\phi$ efficiency, particularly in sparse reward settings.
>
> ----------
> Reference
>
> (Zhang et al., 2021) Amy Zhang et al. Learning invariant representations for reinforcement learning without reconstruction. ICLR. 2021.
>
> (Castro et al., 2021) Pablo Samuel Castro et al. MICo: Improved representations via sampling-based state similarity for markov decision processes. NeurIPS. 2021.
>
> (Castro, 2020) Pablo Samuel Castro. Scalable methods for computing state similarity in deterministic markov decision processes. AAAI. 2020.
>
> (Ferns & Precup, 2014) Norm Ferns and Doina Precup. Bisimulation metrics are optimal value functions. UAI. 2014.

---

### Meta-Review · Area_Chair_AyEo · 2023-12-10

**Metareview:**

The paper is concerned with representation learning in RL. In particular, it builds upon bisimulation metric based representation learning and attempts to integrate long horizon supervision signals that capture relationships among states further apart, this arguably learning a more informative representation.

All reviewers find the high level motivation of the paper sound, and agree that the proposed enhancements of the bisimulation metric are meaningful.

However, the majority of the reviewers find the paper writing and the method exposition quite confusing and at times erroneous. To name a few examples, a reviewer finds issues with the ambitious motivation of the approach that exaggerates the contributions of the paper, and also does not give justice to bisimulation iterative single step update as means to capture long-term dependences (reviewer JyET and w1s3); there are issues with soundness of Theorems 3 and 5 (reviewer JyET); the paper contains many errors (see review w1s3). Further, a reviewer points out that experiments aren't convicing / strong enough , and ideally the method should be evaluated on other environments.

**Justification For Why Not Higher Score:**

The paper received 2 x reject, 1 x borderline reject and 1 x accept.

Although the reviewers appreciate the overall idea and results in the paper, they all agree and feel that the paper is premature for publication. In particular, all reviewers find issues with overly exaggerated statements, un-refined motivation, confusing / erroneous theory, etc. Some reviewers find issues with the presented results, in particular the fact that it is evaluated on a single environment. The concern re writing and exposition are also brought forward even in the accept review.

Although the authors do try to address the above concerns in the rebuttal, by incorporating / clarifying many of the writing suggestions, as well as running experiments on a second environment, Meta-World, the ACs feel that the paper needs to go through more work before it can be accepted. Hence, it is rejected from ICLR 2024.

**Justification For Why Not Lower Score:**

n/a

---

### Decision · Program_Chairs · 2024-01-16

Reject